# GenCue: Generation-Oriented Video Captions for High-Fidelity Text-to-Video

## Abstract

The performance of text-to-video (T2V) models critically depends on the quality of their training captions. However, captions produced by current multimodal large language models (MLLMs) often lack fine-grained visual grounding, temporal coverage, and cinematic expressiveness, limiting models' ability to accurately follow instructions and reconstruct details and dynamics. We present GenCue, a generation-oriented video captioning framework designed to close this gap and enable high-fidelity T2V training. We first introduce the GenCue-SFT-1M and GenCue-RL-8k data suite: the former is a large-scale, schema-guided corpus aided by specialized expert models, while the latter is a carefully curated, high-quality subset providing precise supervision signals for post-training. Building on this data foundation, we propose a reinforcement learning paradigm with a checklist-based reward that explicitly evaluates key generation dimensions. We further introduce Reference-Augmented GRPO and a prefix-sharing rollout strategy, which together enable effective and efficient long-context optimization. Experiments on T2V captioning and reconstruction benchmarks demonstrate that GenCue significantly outperforms prior approaches, yielding substantial improvements in object coverage, temporal coherence, and cinematic quality.

## 1 Introduction

Text-to-video (T2V) generation (Zheng et al., 2024; Yang et al., 2024b; Ma et al., 2025; HaCohen et al., 2024) has advanced rapidly with diffusion transformers trained on large-scale multimodal corpora, enabling the synthesis of realistic and temporally coherent videos from natural-language prompts. Beyond entertainment, these systems are increasingly used by both everyday users and professional creators. In all cases, the quality of the learned alignment between videos and their textual descriptions is pivotal: captions that are precise, complete, and visually grounded reduce hallucination, improve instruction following, and enable faithful rendering of fine detail and scene dynamics.

The prevailing T2V models rely on recaptioned datasets produced automatically by multimodal large language models (MLLMs) (Liu et al., 2023; Bai et al., 2025; Chen et al., 2024b; Liu et al., 2025). While this strategy has unlocked scale and fueled recent progress, existing captions remain far from ideal. Even leading T2V models struggle with subtle yet critical aspects of video fidelity: **(1) Detail accuracy and coverage.** Fine-grained visual and temporal cues—such as small objects, attributes, motions, and scene transitions—are often under-described or omitted, limiting the model's ability to faithfully reconstruct them. **(2) Cinematic expressiveness.** Key aspects of cinematography—such as shot types and composition, lens choices, camera movements, and lighting styles—are often overlooked, resulting in videos that appear visually flat and fail to meet the needs of professional content creators.

To address these challenges, we introduce **GenCue**, a framework for generating high-fidelity, generation-oriented video descriptions tailored for T2V training. Our approach improves both the data and training dimensions of caption modeling.

On the data side, grounding learning in faithful, cinematic evidence, GenCue leverages two purpose-built datasets. *GenCue-SFT-1M* scales supervised fine-tuning with a schema-guided caption synthesis pipeline: an MLLM drafts structured JSON; specialized experts for shot type/angle/position,

Figure 1: Qualitative comparison of captions from Qwen2.5-VL-7B (left) and our GenCue (right). GenCue's output more closely matches the video, capturing finer object details, camera motions, and temporal dynamics. Red text marks incorrect or hallucinated elements, while green highlights accurate, detailed, and visually grounded descriptions.

camera motion, and scene layout verify and complete fields; a fusion stage distills them into concise, generation-ready prose. For post-training, *GenCue-RL-8k* then offers a balanced, high-quality subset with content-focused checklist vetted by strong vision models and human annotators, supplying precise supervision signals that enhance caption fidelity and coverage.

On the learning side, we enhance caption training with novel reinforcement learning techniques. First, we introduce a checklist-based reward that evaluates captions across scene content, temporal dynamics, visual presentation, and cinematography, encouraging faithful coverage and concise, generation-ready descriptions. Second, we propose Reference-Augmented GRPO (*RefGRPO*) with reference-anchored group rollouts, prodiving guidance that accelerates learning on complex samples and helps the model go beyond its prior knowledge. To scale reinforcement learning to long video contexts, we further introduce a prefix-sharing rollout strategy that reuses computation across rollout variants with the same video prefix, achieving up to 90% reduction in FLOPs and 60% reduction in memory usage.

Extensive experiments on T2V-oriented video captioning and text-to-video reconstruction tasks demonstrate that GenCue achieves state-of-the-art performance, substantially enhancing caption–video fidelity (as is shown in Figure 1). Our model delivers more faithful object coverage, richer temporal dynamics, and stronger cinematographic expressiveness. We will release all datasets, models, and evaluation tools to foster progress in the community. We hope that GenCue will serve as a foundation infrastructure for developing controllable, creative, and reliable T2V systems.

## 2 RELATED WORK

**Text-to-Video Generation.** Recent advances in text-to-video (T2V) generation have produced high-quality models (Polyak et al., 2024; Wang et al., 2024d; Ma et al., 2024; Chen et al., 2024a; Wang et al., 2024a) capable of generating plausible videos from simple prompts. However, these models often struggle with complex instructions that require precise instance-level details, intricate camera movements, or coherent temporal dynamics. Analysis of existing video-text datasets suggests that such limitations largely stem from suboptimal training captions, which frequently omit fine-grained object information, scene layout, or motion cues (Yang et al., 2024a; Ju et al., 2024; Chen et al., 2024c). Traditional recaptioning methods, whether manual (Zhou et al., 2018; Chen &

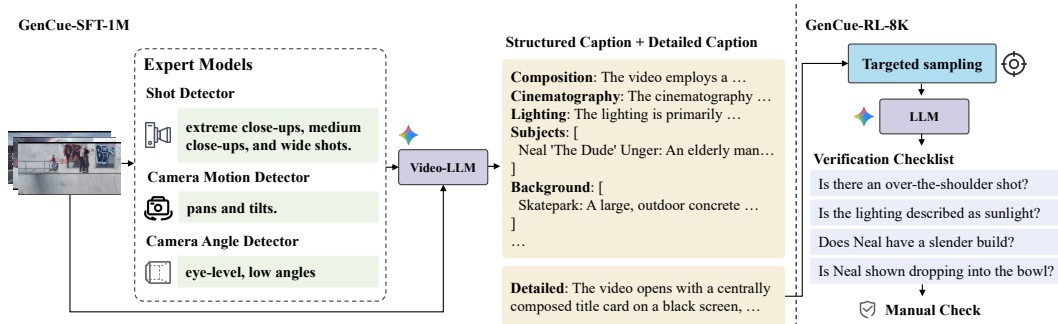

Figure 2: Overall data pipeline of GenCue-SFT-1M and GenCue-RL-8k. GenCue-SFT-1M is constructed with schema-guided structured JSON representations, aided by specialized expert models for shot type, camera motion, and angle, then distilled into long, fluent captions for training. GenCue-RL-8k features targeted clip sampling, manual refinement, and automatically derived checklist points to enable reward-based optimization.

Dolan, 2011) or automatic (Wang et al., 2023), have not consistently captured this level of detail. For T2V generation, therefore, accurate and comprehensive captions are critical, as they provide the necessary guidance for models to reproduce detailed visual content and coherent motion patterns.

**Video Captioning.** Video captioning generates natural language descriptions of videos, supporting tasks such as understanding (Doveh et al., 2023), retrieval (Ma et al., 2024), and motion control (Wang et al., 2024d). Accurate captions are crucial for text-to-video (T2V) generation (Polyak et al., 2024). Early free-form methods (Chen et al., 2024a; Wang et al., 2024a) lacked controllability and omitted details. Structured datasets like MiraData (Ju et al., 2024), VDC (Chai et al., 2024), and Vript (Yang et al., 2024a) emphasized subjects, backgrounds, or cinematography, while event-centric approaches (Wang et al., 2024b; He et al., 2024) captured temporal dynamics for coherent narratives. Scaling relies on video recaptioning: manual annotations (Zhou et al., 2018; Chen & Dolan, 2011; Wang et al., 2020) are accurate but limited, whereas MLLMs enable automatic recaptioning. Short-caption datasets (Chen et al., 2024a; Wang et al., 2023) improve efficiency but reduce detail, while dense-caption corpora (Nan et al., 2024; Yang et al., 2024a; Chen et al., 2024c) enrich content at the cost of hallucination or redundancy. Structured designs like MiraData (Ju et al., 2024) help, yet fine-grained detail, coherence, and cinematic expressiveness remain challenging, limiting high-fidelity T2V generation (Chen et al., 2025).

## 3 METHODS

### 3.1 HIGH-FIDELITY CAPTION DATASET CONSTRUCTION

#### 3.1.1 GENCUE-SFT-1M DATASET

To obtain high-fidelity supervision for captioning model training, we construct a large-scale caption dataset, GenCue-SFT-1M, using a schema-guided pipeline designed to capture the diverse aspects of video content with high coverage and controllability (Figure 2). We first employ a suite of expert models trained on proprietary annotated multi-dimensional labels. These models include classifiers for shot type, camera motion, and camera angle. They focus on cinematography-related attributes that are challenging for general-purpose multimodal language models to recognize consistently.

Conditioned on the video and these expert-derived signals, a MLLM (Gemini-2.5-Flash) is prompted to produce a structured JSON description that decomposes the video into complementary dimensions. The schema is designed to be comprehensive and modular, covering: **(1) Scene Content:** background elements, primary subjects, and their spatial relationships; **(2) Temporal Dynamics:** event narration, motion analysis, dynamic changes, and scene transitions. **(3) Visual Presentation:** element arrangement, dominant color scheme, lighting configuration, and visual style; **(4) Cinematography:** shot type, angle, position, and camera movement, abstracted from specific video content;

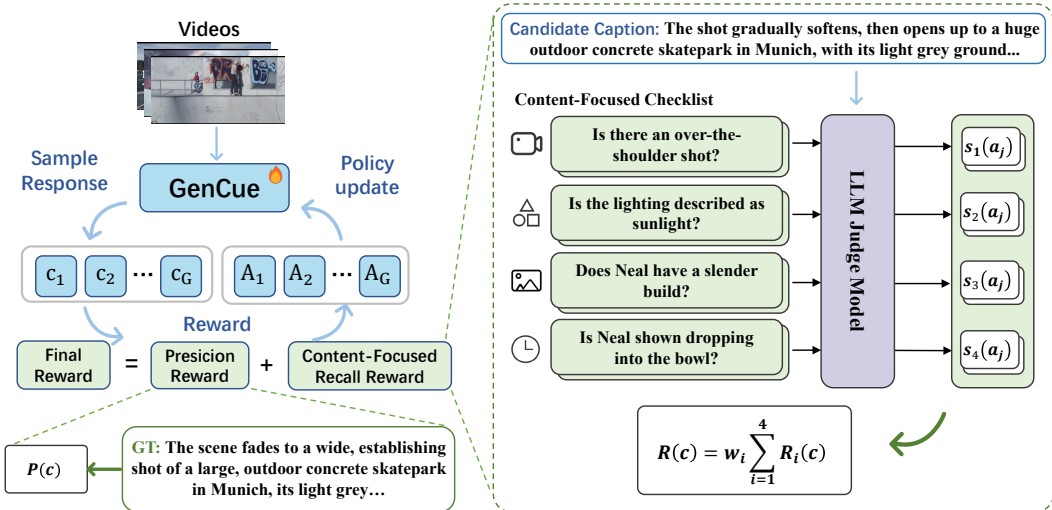

Figure 3: Reinforcement learning framework of GenCue. Our approach optimizes the captioning policy with a content-focused checklist reward that explicitly evaluates scene content, temporal dynamics, visual presentation, and cinematography. We further employ Reference-Augmented GRPO to guide learning with reference-anchored rollouts and adopt a prefix-sharing strategy to efficiently scale RL to long video contexts.

The structured representation decomposes complex scenes into atomic components and lists, enabling detailed coverage of small objects, fine-grained actions, and subtle cinematic cues that are often missing from conventional captions. It also allows targeted updates when stronger expert models become available, without perturbing other dimensions. Finally, the JSON is summarized into fluent natural-language captions by the MLLM, yielding long, coherent descriptions suitable for training. The resulting 1M dataset exhibits substantially greater semantic density and cinematographic expressiveness than standard video caption corpora, providing stronger supervision for both content fidelity and stylistic control. Detailed statistics of the constructed dataset are provided in Appendix D.

### 3.1.2 GENCUE-RL-8K DATASET

To support reinforcement learning, we construct GenCue-RL-8k, a curated set of 8k video–caption pairs specifically tailored for reward-based optimization.

The construction pipeline consists of three stages: **(1) Targeted sampling.** We select clips from a large-scale corpus with controlled distributions of shot types, camera angles, motions, and scene compositions, ensuring diverse and representative coverage, including challenging cases. **(2) Caption refinement.** Each clip is captioned using the same schema-guided process as GenCue-SFT-1M but with a stronger Gemini-Pro model, then manually verified and edited to yield more detailed and reliable descriptions. **(3) Verification point generation.** For each refined caption, we automatically derive a structured set of verification points spanning four generation-relevant dimensions (objects, attributes, motions, and cinematography). Each point corresponds to a YES/NO query checking whether the candidate caption correctly captures that element. On average, a caption yields about 80 verification points. This fine-grained checklist enables the reward model to provide objective and reproducible scores for caption rollouts during RL training.

The resulting dataset provides diverse scenarios, precise reference captions, and dense supervision signals, making it a compact yet effective foundation for reinforcement learning.

### 3.2 GENCUE MODEL

Our GenCue model training pipeline is designed to progressively endow a base MLLM (e.g., Qwen-VL-7B) with stronger video–text alignment and generation-oriented captioning ability. We first

perform supervised fine-tuning (SFT) on the large-scale *GenCue-SFT-1M* dataset, which teaches the model to produce fine-grained, cinematography-aware captions. We then apply reinforcement learning (RL) on *GenCue-RL-8k* to directly optimize caption–video fidelity. Our RL stage introduces three key innovations: a content-focused checklist reward to directly align captions with verifiable visual evidence, a reference-guided GRPO to stabilize learning and encourage improvement on challenging cases, and a PrefixGrouper strategy that makes group rollouts on long video contexts computationally scalable. Together, these steps yield a captioner that is more faithful, temporally precise, and cinematographically expressive, ultimately serving as a key enabler for scalable construction high-fidelity, generation-oriented training corpora for text-to-video models.

### 3.2.1 REWARDING CAPTIONS THROUGH CONTENT-FOCUSED CHECKLIST

Evaluating video captions remains challenging because most existing metrics—such as BLEU, CIDEr, or CLIP-based similarity—focus on surface-level lexical overlap or coarse semantic alignment. They rarely assess generation-oriented aspects like fine-grained object coverage, temporal coherence, and cinematographic style, which are crucial for high-fidelity T2V generation.

To address this gap, we introduce a content-focused reward framework built on the verification checklist derived from GenCue-RL-8k (Figure 3), providing fine-grained and reproducible supervision tailored for T2V training.

We decompose the evaluation into four generation-relevant dimensions—scene content, temporal dynamics, visual presentation, and cinematography—each represented by a dense set of verification points. Together, these points form a comprehensive checklist that specifies the key elements a faithful caption should capture. During training, an expert LLM acts as a judge by evaluating a candidate caption against each verification point and returning a YES/NO/Partial decision. Each decision is mapped to a scalar score $s_i(a)$, and dimension-level recall $R_i(c)$ is obtained by averaging over all $N_i$ points in dimension $i$. The overall reward $R(c)$ is then computed as a weighted sum across the four dimensions:

$$s_i(a) = \begin{cases} 1.0 & a = \text{YES} \\ 0.5 & a = \text{Partially YES} \\ 0.0 & a \in \{\text{NO, Unknown}\} \end{cases}, \quad R_i(c) = \frac{1}{N_i}\sum_{j=1}^{N_i} s_i(a_j), \quad R(c) = \sum_{i=1}^{4} w_i R_i(c), \quad (1)$$

where $w_i$ denotes the weight assigned to dimension $i$. On average, each video contributes roughly 80 verification points, yielding dense and structured supervision that enforces semantic coverage, temporal coherence, and cinematic fidelity.

Thanks to our PrefixGrouper rollout strategy, which reuses shared video-prefix representations across verification points, these rewards can be computed efficiently and in parallel within a single forward pass. This reformulates caption evaluation as a structured content recall problem, yielding objective, reproducible, and dimension-aware reward signals.

To prevent reward hacking through verbose or hallucinated captions, we introduce a precision component. The same LLM judge is prompted with both ground-truth and predicted captions, and outputs a score $P(c) \in [1, 10]$ indicating the degree of extraneous or irrelevant content. The recall and precision signals are combined into a single judge score:

$$S_{\text{judge}}(c) = \lambda R(c) + (1 - \lambda)P(c), \quad (2)$$

and modulated with a soft length regularization term $\tau(c)$ to encourage conciseness:

$$r(c) = S_{\text{judge}}(c) \cdot \tau(c). \quad (3)$$

This two-level reward structure (recall + precision) ensures captions are both comprehensive and concise, covering critical visual, temporal, and cinematic aspects without over-describing. Empirically, this yields captions with richer cinematographic cues, more accurate temporal narratives, and higher object fidelity, leading to improved T2V generation quality.

### 3.2.2 REFGRPO: IMPROVING GRPO WITH CONDITIONAL REFERENCE GUIDANCE

Group Relative Policy Optimization (GRPO) compares multiple rollouts for the same input and uses their relative rewards to compute advantages, avoiding the need for a learned value function.

However, in T2V captioning, GRPO suffers when (i) all rollouts underperform, yielding nearly zero relative advantages and no useful gradient signal, and (ii) samples require knowledge beyond the model's prior, such as subtle camera motions, where exploration rarely discovers a correct caption. This limits the model's ability to improve on the most challenging cases.

To address this, we augment GRPO with a high-quality reference caption (i.e., the ground-truth caption from GenCue-RL-8k) that is conditionally incorporated into both advantage computation and policy update. For each video sample, the model generates $G$ rollouts $\mathcal{C} = \{c_1, \ldots, c_G\}$, each with reward $r(c_j)$. We track an exponential moving average of batch rewards, $E_t = \gamma E_{t-1} + (1 - \gamma)\bar{r}_t$, where $\bar{r}_t$ is the mean reward at training step $t$. When all rollouts satisfy $\max_j r(c_j) < E_t$, indicating that the model fails to meet its expected performance, we add a reference caption $c_{\text{ref}}$ into the group for advantage calculation and policy update. Its reward is set close to the group mean:

$$r(c_{\text{ref}}) = \min\{r(c_j) \mid r(c_j) > \bar{r}_{\mathcal{C}}\}, \quad \bar{r}_{\mathcal{C}} = \sum_{k=1}^{G} r(c_k)/G. \tag{4}$$

This design treats the reference as a soft performance anchor, refining advantage estimation without causing large shifts in the advantages of the original rollouts. Crucially, it provides a consistent positive signal that gently steers the policy toward reference-like captions.

In the policy update, the reference is handled exactly like another trajectory in the group: we compute its advantage against the expanded set and apply the same GRPO objective as for model rollouts. By supplying guidance only when all rollouts fail, RefGRPO delivers a targeted corrective signal, stabilizing training and improving data efficiency.

### 3.2.3 Context-Sharing Acceleration for Video RL

A key obstacle in GRPO for video captioning is the high cost of repeatedly encoding long video-text prefixes when performing group rollouts. Large group sizes are essential for reducing advantage variance, but conventional GRPO implementations redundantly reprocess the same context for every response candidate, leading to rapidly growing compute and memory usage as the group size scales.

To address this, we propose **PrefixGrouper**, a lightweight algorithm that decouples the expensive context encoding pass from the lightweight generation stage. Instead of duplicating context tokens across all $G$ rollout branches, PrefixGrouper computes the context representation once, then performs grouped attention that reuses this context while attending independently to each output branch. This design preserves exact equivalence with the standard GRPO policy update but reduces computation on the context by up to $\frac{1}{G}$ in the long-context regime.

Practically, PrefixGrouper enables training with larger rollout groups under the same compute budget, which improves gradient estimates and accelerates training. Beyond rollouts, we also apply it to group-based checklist reward computation. Together, these gains make reinforcement learning for video-grounded generation far more efficient and scalable. Full algorithmic details and formal proofs are provided in Appendix B.

## 4 Experiments

### 4.1 Implementation Details

We conducted our experiments on two models, Qwen2.5-VL-3B (Bai et al., 2025) and Qwen2.5-VL-7B (Bai et al., 2025), following a two-stage process: an initial fine-tuning stage and a second reinforcement post-training stage.

In the first stage, to enhance the models' general-purpose capabilities, we constructed a training set by combining our GenCue-SFT-1M data with 200K instruction and caption samples from llava-video-178K (Zhang et al., 2024b), along with 200K caption samples from the Tarsier-585K dataset (Wang et al., 2024b). In the second stage, we trained the models for one epoch using our GenCue-RL-8K dataset.

The input video resolution was set to a minimum of 50,176 pixels, with a minimum of 8 input frames and a base sampling rate of 1 fps. For the fine-tuning stage, we utilized 24 NVIDIA A100 GPUs for the 3B model and 32 NVIDIA A100 GPUs for the 7B model. During the reinforcement post-

Table 1: Evaluation results on VidCapBench-AE video captioning benchmark.

| Models | Aesthetics | Content | Motion | PhysicalLaws | Overall |
|---|---|---|---|---|---|
| *Proprietary Models* | | | | | |
| GPT-4o | 14.1 | 17.5 | 10.2 | 27.9 | 16.8 |
| Gemini-1.5-Pro-002 | 16.4 | 16.9 | 9.8 | 28.4 | 17.1 |
| *Open-source Models (>10B)* | | | | | |
| Tarsier-34B (Wang et al., 2024b) | 14.7 | 12.4 | _7.1_ | 28.1 | 13.5 |
| Qwen2-VL-72B (Wang et al., 2024c) | 12.0 | 11.5 | 5.8 | 27.1 | 12.2 |
| *Open-Source Models (<10B)* | | | | | |
| InternVL2-8B (Chen et al., 2024d) | 9.1 | 10.0 | 4.4 | 23.6 | 10.2 |
| Llava-Next-Video-7B (Zhang et al., 2024a) | 11.3 | 9.6 | 4.4 | 24.4 | 10.6 |
| Qwen2-VL-7B (Wang et al., 2024c) | 12.4 | 9.9 | 4.0 | 26.1 | 11.1 |
| Qwen2.5-VL-3B (Bai et al., 2025) | 9.8 | 9.7 | 4.9 | 26.4 | 10.3 |
| Qwen2.5-VL-7B (Bai et al., 2025) | 13.9 | 13.2 | 6.7 | 26.8 | 13.8 |
| SkyCaptioner-7B (Qiu et al., 2025) | 13.7 | _14.4_ | 6.7 | 28.1 | 14.6 |
| **GenCue-3B (Ours)** | _15.5_ | 13.8 | 6.7 | **31.1** | _14.8_ |
| **GenCue-7B (Ours)** | **16.1** | **15.8** | **7.6** | _29.4_ | **16.2** |

training stage, both the 3B and 7B models were trained on 8 NVIDIA A100 GPUs. More details are provided in the Appendix A.

We evaluated our model on three captioning benchmarks. Among them, VidCapBench (Chen et al., 2025) is a benchmark specifically constructed for the text-to-video domain, covering evaluations from four perspectives: video aesthetics, video content, video motion, and physical laws. The Video Detailed Captions (VDC) (Chai et al.) dataset contains high-quality videos spanning various categories, with evaluation dimensions that include key elements such as camera movement, primary subjects, and background. Lastly, CaReBench (Xu et al., 2024) approaches evaluation from the perspectives of objects and actions, designing precision and recall scores to assess caption quality. Collectively, these benchmarks evaluate the quality of captions across a spectrum of aspects, from subject and background to professional cinematography and style. They effectively reflect the comprehensiveness, professionalism, and accuracy of the generated captions.

## 4.2 RESULTS ON CAPTIONING BENCHMARKS

**Results on VidCapBench** As demonstrated in Table 1, our GenCue-7B and GenCue-3B achieve the highest and the second highest overall accuracy (16.2 and 14.8) among open-source models, consistently outperforming strong prior models across all four professional dimensions: video aesthetics, video content, video motion, and physical laws. This broad superiority highlights its enhanced ability to capture stylistic nuances, core content, complex temporal dynamics, and grounded, realistic scene properties. The results confirm that our approach yields more comprehensive and accurate video descriptions, which are essential for high-fidelity text-to-video generation tasks.

**Results on VDC and CaReBench** We further evaluated our GenCue models' specialized capabilities on the VDC and CaReBench benchmarks, as shown in Table 2. On VDC, GenCue-7B shows a commanding lead in categories vital for generation, achieving the highest scores for both Camera (58.5) and Detailed (56.1) captions. This underscores its superior ability to describe complex cinematography and fine-grained visual details. This proficiency extends to CaReBench, where our model attains the best F1 scores for both the Object (34.3) and Action (31.5) categories, reinforcing its robust capability to accurately narrate key subjects and their interactions. Furthermore, our GenCue-3B also achieves competitive performance even compared with other 7B models.

## 4.3 RESULTS ON TEXT-TO-VIDEO GENERATION

The quality of video detailed captions is intrinsically linked to the performance of text-to-video (T2V) generation tasks. Therefore, following the experimental setup of VidCapBench, we generate detailed captions on this dataset using Qwen2.5-VL-7B, SkyCaptioner-7B, and our GenCue models. Subsequently, we employ two T2V models, Wan2.1 (Wan et al., 2025) and HunyuanVideo (Kong

Table 2: Evaluation results on VDC and CaReBench video captioning benchmarks.

| Models | VDC | | | | CaReBench | |
| --- | --- | --- | --- | --- | --- | --- |
| | Camera | Object | Detailed | Overall | Action | Object |
| *Proprietary Models* | | | | | | |
| Gemini-1.5 Pro | 38.7 | 47.3 | 43.1 | 41.7 | - | - |
| GPT-4o mini | - | - | - | - | 36.8 | 33.8 |
| *Open-source Models* | | | | | | |
| InternVL-2-8B (Chen et al., 2024d) | 39.1 | 44.2 | 33.0 | 37.5 | 23.3 | 22.9 |
| Tarsier-7B (Wang et al., 2024b) | - | - | - | 38.3 | 27.1 | 31.1 |
| Qwen2-VL-7B (Wang et al., 2024c) | 39.0 | 47.8 | 42.5 | 41.6 | 28.8 | 24.0 |
| Qwen2-VL-72B (Wang et al., 2024c) | - | - | - | - | 30.5 | 24.2 |
| Qwen2.5-VL-3B (Bai et al., 2025) | 45.5 | 47.8 | 51.6 | 45.9 | 28.2 | 27.7 |
| Qwen2.5-VL-7B (Bai et al., 2025) | 48.0 | 52.5 | 53.3 | 50.3 | 30.8 | 29.8 |
| SkyCaptioner-7B (Qiu et al., 2025) | 52.7 | 55.4 | 54.9 | 52.6 | 31.2 | 32.2 |
| **GenCue-3B (Ours)** | 53.5 | 56.8 | 54.2 | 51.5 | **31.5** | 32.6 |
| **GenCue-7B (Ours)** | **58.5** | **58.5** | **56.1** | **54.9** | **31.5** | **34.3** |

Table 3: Text-to-video evaluation results using Wan2.1 and HunyuanVideo.

| Methods | Wan2.1 | | | HunyuanVideo | | |
| --- | --- | --- | --- | --- | --- | --- |
| | CLIP ↑ | FVD ↓ | LPIPS ↓ | CLIP ↑ | FVD ↓ | LPIPS ↓ |
| Qwen2.5-VL-7B (Bai et al., 2025) | 0.794 | 484.31 | 0.728 | 0.818 | 539.46 | 0.738 |
| SkyCaptioner-7B (Qiu et al., 2025) | 0.785 | 461.20 | 0.716 | 0.823 | 494.99 | 0.722 |
| **GenCue-3B (Ours)** | 0.807 | 457.56 | 0.712 | 0.829 | 477.30 | 0.722 |
| **GenCue-7B (Ours)** | **0.821** | **423.61** | **0.699** | **0.831** | **454.60** | **0.710** |

et al., 2024), to generate videos based on these captions. The results, as presented in Table 3, demonstrate that our GenCue-3B and GenCue-7B models outperform the competing models across three text-to-video evaluation metrics, including CLIP similarity (Radford et al., 2021), LPIPS (Zhang et al., 2018), and FVD (Unterthiner et al., 2019). This outcome fully showcases the fine-grained detail and professional quality of the captions generated by GenCue for the T2V domain. Further qualitative analysis is provided in Appendix C.1.

## 4.4 Ablation Studies

**Training Strategy Ablation** To validate the effectiveness of our proposed 2-stage training strategy, we evaluated the results under different settings, presented in Table 4. Starting from a baseline score of 45.9 on VDC and 10.3 on VidCapBench, we observe that either the Supervised Fine-Tuning (SFT) stage or the Reinforcement Learning (RL) stage individually brings notable improvements. For instance, RL alone lifts the scores to 48.9 (VDC) and 13.1 (VidCapBench). However, the full two-stage training strategy (+SFT & RL) yields the most significant gains, boosting performance to a final score of 51.5 on VDC and 14.8 on VidCapBench. This demonstrates the strong complementary benefits of both stages in achieving superior captioning quality.

**RL Design Ablation** Furthermore, we dissected our RL design to assess the contribution of each component, with results shown in Table 5. The analysis clearly indicates that removing any of our proposed elements leads to a discernible drop in performance. Notably, the removal of the reference guidance in RefGRPO (w/o Ref) results in the most substantial performance degradation, causing the VDC score to drop by 2.5 points from 51.5 to 49.0. Other components are also crucial; for example, removing the soft performance anchor (w/o Soft, i.e., Eq. 4) lowers the VDC score to 49.9, while removing the precision score (w/o Precision) lowers the VDC score to 50.3. These results underscore the critical and integral role of each design choice, especially RefGRPO, in guiding the policy toward higher-quality captions.

Table 4: Training Strategy Ablations. Comparison of baseline MLLM, SFT-only, RL-only, and combined SFT+RL training strategies on VDC and VidCapBench..

| SFT | RL | VDC | VidCapBench |
|---|---|---|---|
| ✗ | ✗ | 45.9 | 10.3 |
| ✓ | ✗ | 47.3 | 12.4 |
| ✗ | ✓ | 48.9 | 13.1 |
| ✓ | ✓ | **51.5** | **14.8** |

Table 5: Ablation of RL Components. Comparison of the full GenCue RL framework with variants removing RefGRPO, soft anchor, or precision reward term.

| Methods | VDC | VidCapBench |
|---|---|---|
| w/ all | **51.5** | **14.8** |
| w/o Ref | 49.0 | 13.9 |
| w/o Soft | 49.9 | 14.2 |
| w/o Precision | 50.3 | 14.0 |

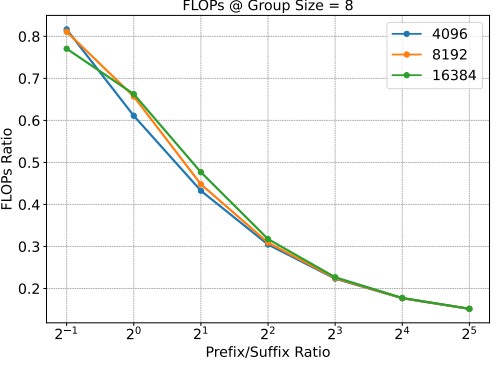

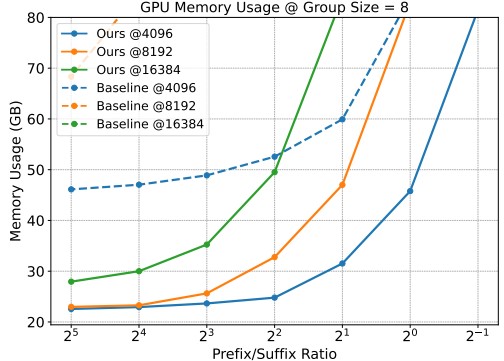

(a) FLOPs ratio between PrefixGrouper and standard GRPO as the input length grows.

(b) GPU memory usage for both methods across different total input lengths.

Figure 4: Efficiency of the proposed PrefixGrouper. FLOPs and GPU memory comparison under group size 8 and varying prefix-to-suffix ratios.

## 4.5 EFFICIENCY

In this section, we conduct a further investigation into the reductions in FLOPs and GPU memory afforded by PrefixGrouper, with the results illustrated in Figure 4. As shown in Figure 4(a), for a group size of 8 and across various prefix input lengths, the ratio of computational load between the standard GRPO algorithm and PrefixGrouper drops below 0.2 as the prefix-to-suffix length ratio increases, corresponding to more than 80% reduction in FLOPs. This value corroborates our theoretical derivation that the computational load can be reduced to as low as $\frac{1}{G}$. Additionally, Figure 4(b) demonstrates that PrefixGrouper maintains a significantly lower memory footprint across different prefix lengths and prefix-to-suffix length ratios. This characteristic enables the model to support GRPO training with longer contexts on the same hardware. Further performance analysis with different group sizes is available in Appendix C.2.

## 5 CONCLUSION

In this work, we tackled a key limitation in text-to-video training: the lack of captions that capture both fine-grained visual details and cinematographic context. We introduced GenCue, a unified framework that addresses this challenge through innovations in data and learning. By introducing the large-scale, schema-guided GenCue-SFT-1M dataset and the curated, high-quality GenCue-RL-8k supervision set, we deliver captions with stronger visual grounding, temporal coverage, and cinematic richness. Our checklist-based reward, combined with Reference-Augmented GRPO and prefix-sharing rollouts, enables efficient long-context reinforcement learning and substantially boosts caption–video fidelity. Extensive experiments on T2V captioning and reconstruction tasks confirm that GenCue improves object coverage, temporal coherence, and cinematographic expressiveness over prior approaches. We believe GenCue can serve as a practical building block for improving future T2V training pipelines and enabling high-fidelity and expressive video generation.

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

# APPENDIX

## A    IMPLEMENTATION DETAILS

Table 6 summarizes the hyperparameters used across different training stages. During the reinforcement learning (RL) training process (Stage 2), we use a group size of 8 and leverage vLLM (Kwon et al., 2023) to accelerate model rollouts. To maximize GPU utilization, a dynamic offloading strategy is employed for the judge, rollout, and reference models.

Table 6: Hyperparameter settings.

| Hyperparameter | Stage1 | Stage2 |
|---|---|---|
| Batch Size | 128 (3B) / 144 (7B) | 8 |
| Learning Rate (LR) | 1e-6 | 1e-6 |
| LR Schedule | cosine | cosine |
| LR Warmup Ratio | 0.03 | 0.03 |
| Epoch | 1 | 2 |

## B    DETAILED ILLUSTRATION OF PREFIXGROUPER

### B.1    ALGORITHM IMPLEMENTATION

For clarity, we assume a forward batch size of 1, where the input consists of a single query prefix $P \in \mathbb{R}^{1 \times L \times D}$ (typically containing system prompts, multi-modal inputs, and questions). Given a group size $G$, the model samples $G$ response candidates $\{R_1, R_2, \ldots, R_G\}$ from $P$, where each

Figure 5: Method illustration of Grouped Attention in PrefixGrouper.

$R_i \in \mathbb{R}^{1 \times L_i \times D}$. Traditional approaches, which we refer to as Repeated-Prefix Forward, process the inputs as follows:

$$x_i = [P; R_i], X_{\text{base}} = \text{pad}(x_1, x_2, \ldots, x_G), \tag{5}$$

where $\text{pad}$ denotes the length-padding operation, resulting in $X_{\text{base}} \in \mathbb{R}^{G \times (L + \max(L_1, \ldots, L_G)) \times D}$.

In PrefixGrouper, we instead concatenate the shared prefix with all response suffixes along the sequence dimension:

$$X_{\text{ours}} = [P; R_1; R_2; \ldots; R_G], \tag{6}$$

where $X_{\text{ours}} \in \mathbb{R}^{1 \times (L + \sum_{i=1}^{G} L_i) \times D}$. It is obvious that the concatenated samples maintain equivalence with Repeated-Prefix Forward in word embedding and FFN layers.

While Repeated-Prefix Forward directly applies self-attention:

$$O = \text{Attn}(Q, K, V, M), \tag{7}$$

where the $G$ rollouts per prefix lead to redundant computation on the repeated prefix in $X_{\text{base}}$, particularly significant for long prefixes. In contrast, our method decomposes the attention computation into two kernel calls (which we refer to as Grouped Attention):

$$\begin{aligned} O_{\text{prefix}} &= \text{Attn}(Q_{\text{prefix}}, K_{\text{prefix}}, V_{\text{prefix}}, M_{\text{prefix}}), \\ O_{\text{suffix}} &= \text{Attn}(Q_{\text{suffix}}, K_{\text{prefix, suffix}}, V_{\text{prefix, suffix}}, M_{\text{prefix, suffix}}), \\ O &= \text{group}(O_{\text{prefix}}, O_{\text{suffix}}), \end{aligned} \tag{8}$$

where $\text{group}(\cdot, \cdot)$ represents concatenating using index select.

For positional encoding (using RoPE or its variants), we set the position id of each token the same as the corresponding position in the Repeated-Prefix Forward. The complete process of Grouped Attention in PrefixGrouper is summarized in Figure 5 and Algorithm 1.

### B.2 GRADIENT EQUIVALENCE OF PREFIXGROUPER

A critical property of PrefixGrouper is its **theoretical equivalence** to Repeated-Prefix Forward in both forward outputs and backward gradients. Forward equivalence is self-evident: every token undergoes fundamentally identical computations in both algorithms. Gradient equivalence is demonstrated below:

---

Algorithm 1: Pseudocode of Grouped Attention in a PyTorch-like style

---

```python
def grouped_attention(self, q, k, v, prefix_grouper, **kwargs):
    """
    q, k, v: Shape [b, num_heads, seq_len, head_dim]. q, k should be pre-processed with RoPE
        in advance.
    prefix_grouper: A plug-and-play module implemented by us.
    kwargs: Any arguments needed by the attention operation.
    """
    # Split the concatenated samples into prefix and suffix
    q_prefix, k_prefix, v_prefix, q_suffix, k_suffix, v_suffix = prefix_grouper.ungroup(q, k,
        v)
    # Attention call
    prefix_attn_output, _ = attention_interface(
        self,
        q_prefix,
        k_prefix,
        v_prefix,
        # NOTE: Attention mask is pre-computed by prefix_grouper
        prefix_grouper.prefix_attn_mask.to(q_prefix.device),
        **kwargs,
    )
    suffix_attn_output, _ = attention_interface(
        self,
        q_suffix,
        prefix_grouper.batch_repeat_cat(k_prefix, k_suffix, cat_dim=2),
        prefix_grouper.batch_repeat_cat(v_prefix, v_suffix, cat_dim=2),
        # NOTE: Attention mask is pre-computed by prefix_grouper
        prefix_grouper.suffix_attn_mask.to(q_prefix.device),
        **kwargs,
    )
    # Concatenate the prefix and suffix output
    # The input shape should be [b, seq_len, num_heads, head_dim]
    attn_output = prefix_grouper.group(prefix_attn_output, suffix_attn_output)
    return attn_output, None
```

---

`attention_interface`: the attention operation; `prefix_grouper`: implemented using `torch.autograd.Function`.

---

**Lemma B.1 (Gradient Equivalence)**

Under the Grouped Attention in our PrefixGrouper, the gradients of the policy loss function with respect to model parameters $\theta$ are identical to those computed by the original GRPO algorithm, i.e.,

$$\nabla_\theta \mathcal{J}_{\text{ours}}(X_{\text{ours}}, A) \equiv \nabla_\theta \mathcal{J}_{\text{base}}(X_{\text{base}}, A), \tag{9}$$

where $A$ denotes the advantage of the rollout responses.

*Proof.* For clarity, we assume batch size $B = 1$. For a transformer with parameters $\theta$, the total gradient $\nabla_\theta \mathcal{J}$ is the sum of per-token contributions:

$$\nabla_\theta \mathcal{J} = \sum_{t=1}^{T} \underbrace{\frac{\partial \mathcal{J}}{\partial \mathbf{h}_t} \cdot \nabla_\theta \mathbf{h}_t}_{\text{token } t\text{'s gradient contribution}}, \tag{10}$$

where $\mathbf{h}_t$ is the hidden state at position $t$. This holds under additive loss decomposition: $\mathcal{J} = \sum_t \mathcal{J}_t(\mathbf{h}_t)$ (e.g., next-token prediction)

We decompose the transformer architecture into two computational categories: 1. **Attention operations**: $O = \text{Attn}(Q, K, V)$. 2. **Pointwise operations**: MLP, QKV projections, and normalization layers.

All learnable parameters reside in pointwise operations. Gradient equivalence is established by analyzing each category separately.

PART 1: ATTENTION GRADIENT

Consider the attention output $O$ and its gradients $\nabla_{Q,K,V} O$. For suffix tokens ($R_i$), both implementations compute identical gradients due to equivalent computational paths:

$$\nabla_{Q_{\text{suffix}}, K_{\text{suffix}}, V_{\text{suffix}}} O_{\text{ours}} \equiv \nabla_{Q_{\text{suffix}}, K_{\text{suffix}}, V_{\text{suffix}}} O_{\text{base}}. \tag{11}$$

For prefix tokens ($P$), the gradient is calculated as follows:

$$\nabla_{P_i(Q,K,V)}O_{\text{base}} = \underbrace{\nabla_{P_i(Q,K,V)}O_i^{\text{prefix}}}_{\text{prefix-only}} + \underbrace{\nabla_{P_i(Q,K,V)}O_i^{\text{suffix}}}_{\text{response interaction}}, \tag{12}$$

$$\nabla_{P(Q,K,V)}O_{\text{ours}} = \nabla_{P(Q,K,V)}O^{\text{prefix}} + \frac{1}{G}\sum_{i=1}^{G}\left(\nabla_{P(Q,K,V)}O_i^{\text{suffix}}\right). \tag{13}$$

PART 2: POINTWISE OPERATION GRADIENT

For any pointwise operation parameter $\theta$, gradients are computed as:

$$\nabla_{\theta}\mathcal{J}_{\text{base}} = \frac{1}{G}\sum_{i=1}^{G}\left(\sum_{t\in P_i}\frac{\partial\mathcal{J}_{\text{base}}}{\partial\mathbf{h}_t}\nabla_{\theta}\mathbf{h}_t + \sum_{t\in R_i}\frac{\partial\mathcal{J}_{\text{base}}}{\partial\mathbf{h}_t}\nabla_{\theta}\mathbf{h}_t\right), \tag{14}$$

$$\nabla_{\theta}\mathcal{J}_{\text{ours}} = \sum_{t\in P}\frac{\partial\mathcal{J}_{\text{ours}}}{\partial\mathbf{h}_t}\nabla_{\theta}\mathbf{h}_t + \frac{1}{G}\sum_{i=1}^{G}\sum_{t\in R_i}\frac{\partial\mathcal{J}_{\text{ours}}}{\partial\mathbf{h}_t}\nabla_{\theta}\mathbf{h}_t. \tag{15}$$

In the final output, the GRPO loss $\mathcal{J}$ depends *only* on response tokens $R_i$:

$$\forall t\in P,\ \frac{\partial\mathcal{J}}{\partial\mathbf{h}_t} = 0 \quad \text{(both algorithms)}, \tag{16}$$

so based on Eq. 14 and Eq. 15, the gradients of the final FFN and output embedding layers are equivalent in both algorithms.

For the attention part, we have:

$$\frac{1}{G}\sum_{i=1}^{G}\nabla_{P_i(Q,K,V)}O_{\text{base}} = \nabla_{P(Q,K,V)}O_{\text{ours}}, \tag{17}$$

so substitute Eq. 17 into Eq. 14 and Eq. 15, we have:

$$\begin{aligned}
\nabla_{\theta}\mathcal{J}_{\text{base}} &= \frac{1}{G}\sum_{i=1}^{G}\left(\sum_{t\in P_i}\frac{\partial\mathcal{J}_{\text{base}}}{\partial\mathbf{h}_t}\nabla_{\theta}\mathbf{h}_t + \sum_{t\in R_i}\frac{\partial\mathcal{J}_{\text{base}}}{\partial\mathbf{h}_t}\nabla_{\theta}\mathbf{h}_t\right) \\
&= \frac{1}{G}\sum_{i=1}^{G}\sum_{t\in P_i}\frac{\partial\mathcal{J}_{\text{base}}}{\partial\mathbf{h}_t}\nabla_{\theta}\mathbf{h}_t + \frac{1}{G}\sum_{i=1}^{G}\sum_{t\in R_i}\frac{\partial\mathcal{J}_{\text{base}}}{\partial\mathbf{h}_t}\nabla_{\theta}\mathbf{h}_t \\
&= \sum_{t\in P}\frac{\partial\mathcal{J}_{\text{ours}}}{\partial\mathbf{h}_t}\nabla_{\theta}\mathbf{h}_t + \frac{1}{G}\sum_{i=1}^{G}\sum_{t\in R_i}\frac{\partial\mathcal{J}_{\text{ours}}}{\partial\mathbf{h}_t}\nabla_{\theta}\mathbf{h}_t \\
&= \nabla_{\theta}\mathcal{J}_{\text{ours}}.
\end{aligned} \tag{18}$$

Therefore, through layer-by-layer backpropagation, gradient equivalence propagates upstream from output layers. Thus, $\nabla_{\theta}\mathcal{J}_{\text{ours}}(X_{\text{ours}}, A) \equiv \nabla_{\theta}\mathcal{J}_{\text{base}}(X_{\text{base}}, A)$ for all parameters $\theta$. $\qquad\square$

This property ensures that PrefixGrouper's computational efficiency gains come **without trade-offs**, which means it accelerates training and reduces GPU memory while preserving model performance identically.

### B.2.1 COMPUTATIONAL COST ANALYSIS OF PREFIXGROUPER

PrefixGrouper employs the Shared-Prefix Forward approach, which demonstrates increasingly significant advantages over Repeated-Prefix Forward as group size grows:

**Lemma B.2 (Computation Reduction)**

Given a group size $G$ and sequence lengths $L_p$ (prefix), $L_r$ (response), when $L_p \gg L_r$, PrefixGrouper reduces FLOPs to $\frac{1}{G}$ of the Repeated-Prefix Forward approach.

*Proof.* For clarity, we assume batch size $B = 1$ and uniform response length $L_r$ across all $G$ responses. Let $L_p$ denote prefix length, $d$ head dimension, and $n$ number of attention heads. The computational complexity is analyzed separately for causal attention and pointwise operations (MLP & QKV projections).

**Causal Attention Operation:** The baseline Repeated-Prefix Forward method computes:

$$\mathcal{C}_{\text{attn}}^{\text{base}} = G(L_p + L_r)^2 dn \tag{19}$$

Our PrefixGrouper decomposes attention into prefix and suffix components:

$$\mathcal{C}_{\text{attn}}^{\text{ours}} = \underbrace{L_p^2 dn}_{\text{prefix self-attn}} + \underbrace{GL_r(2L_p + L_r)dn}_{\text{suffix attn}} \tag{20}$$

The complexity ratio simplifies to:

$$\frac{\mathcal{C}_{\text{attn}}^{\text{ours}}}{\mathcal{C}_{\text{attn}}^{\text{base}}} = \frac{L_p^2 dn + GL_r(2L_p + L_r)dn}{G(L_p + L_r)^2 dn}$$
$$= \frac{L_p^2 + GL_r(2L_p + L_r)}{G(L_p + L_r)^2} \tag{21}$$

As $L_p \gg L_r$, the asymptotic limit is:

$$\lim_{L_p/L_r \to \infty} \frac{\mathcal{C}_{\text{attn}}^{\text{ours}}}{\mathcal{C}_{\text{attn}}^{\text{base}}} = \frac{L_p^2}{GL_p^2} = \frac{1}{G} \tag{22}$$

**Pointwise Operation:** Let $\mathcal{C}_{\text{ffn}}$ denote FLOPs per token for MLP and projections. The baseline requires:

$$\mathcal{C}_{\text{pointwise}}^{\text{base}} = G(L_p + L_r)\mathcal{C}_{\text{ffn}} \tag{23}$$

while PrefixGrouper computes:

$$\mathcal{C}_{\text{pointwise}}^{\text{ours}} = L_p\mathcal{C}_{\text{ffn}} + GL_r\mathcal{C}_{\text{ffn}} \tag{24}$$

The asymptotic ratio is:

$$\lim_{L_p/L_r \to \infty} \frac{\mathcal{C}_{\text{pointwise}}^{\text{ours}}}{\mathcal{C}_{\text{pointwise}}^{\text{base}}} = \lim_{L_p/L_r \to \infty} \frac{L_p + GL_r}{G(L_p + L_r)} = \frac{1}{G} \tag{25}$$

**Conclusion:** Both attention and pointwise operations exhibit $\mathcal{O}(1/G)$ complexity reduction under $L_p \gg L_r$ conditions. $\square$

Under long-prefix scenarios (e.g., multi-modal inputs or extended text contexts), PrefixGrouper effectively reduces computational load and memory consumption, requiring only marginal computational increase as group size scales, demonstrating its superior efficiency.

## C  MORE EXPERIMENTS

### C.1  QUALITATIVE RESULTS ON TEXT-TO-VIDEO GENERATION

We visualize the video generation results in Figure 6, Figure 7 and Figure 8. The results demonstrate that our GenCue model possesses a superior understanding of cinematography and an enhanced capability for describing fine-grained details. Consequently, it can effectively assist text-to-video models in generating videos of a higher quality.

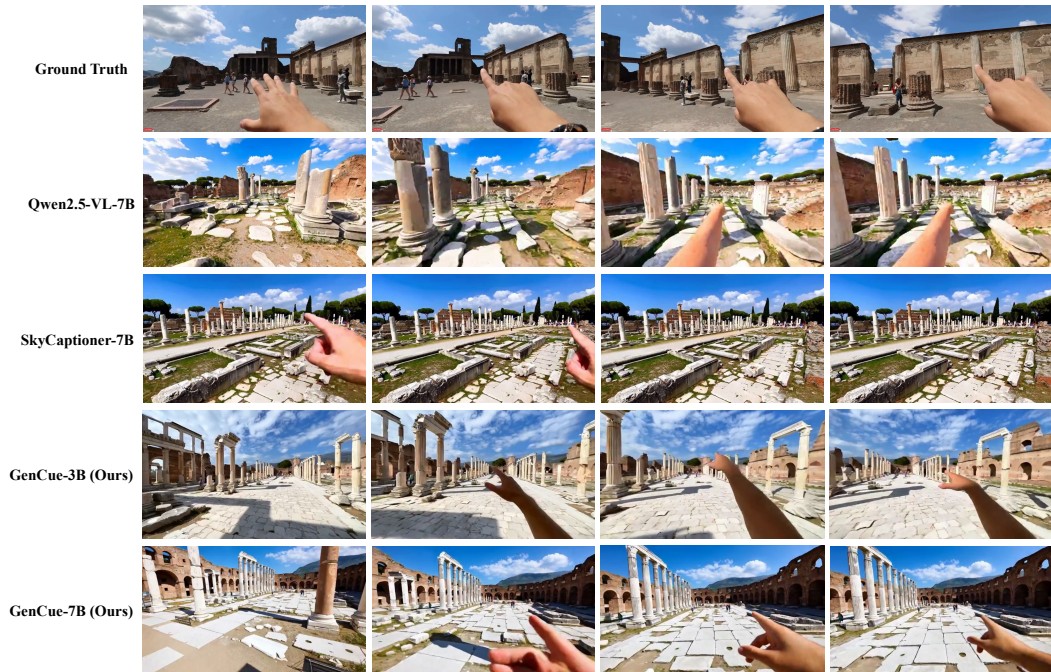

Figure 6: Comparison of reconstructed videos guided through captions generated by different captioning models. Note that our GenCue-3B and GenCue-7B successfully detect the tracking right camera motion, while Qwen2.5-VL-7B describes it as moving forward, and SkyCaptioner-7B predicts a static camera shot.

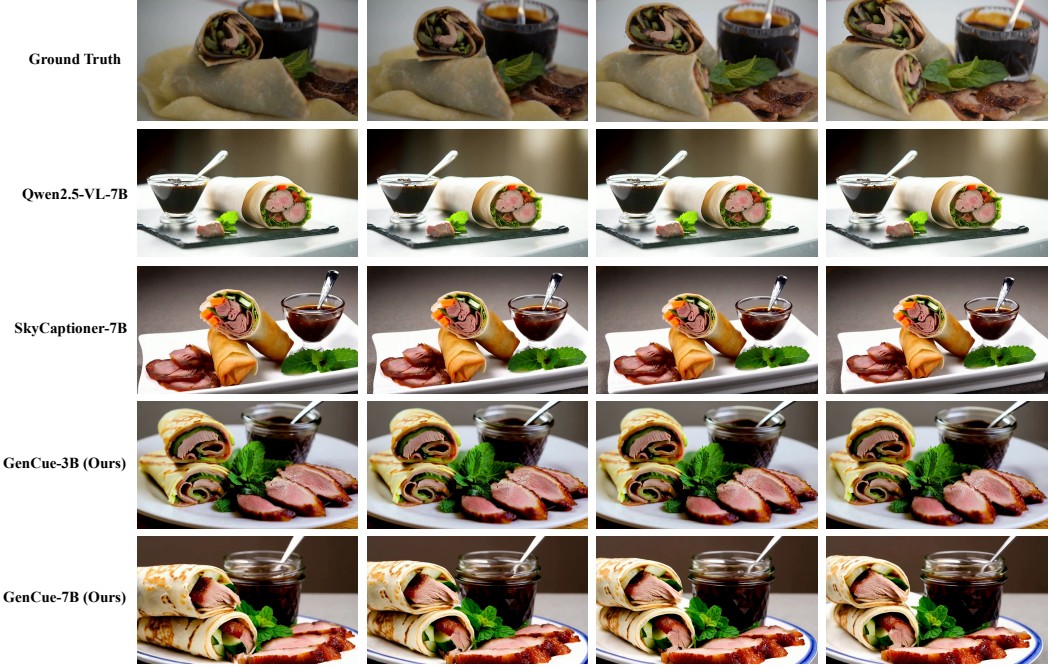

Figure 7: Comparison of reconstructed videos guided through captions generated by different captioning models. Note that our GenCue-3B and GenCue-7B successfully describes the spatial relationships and camera motion.

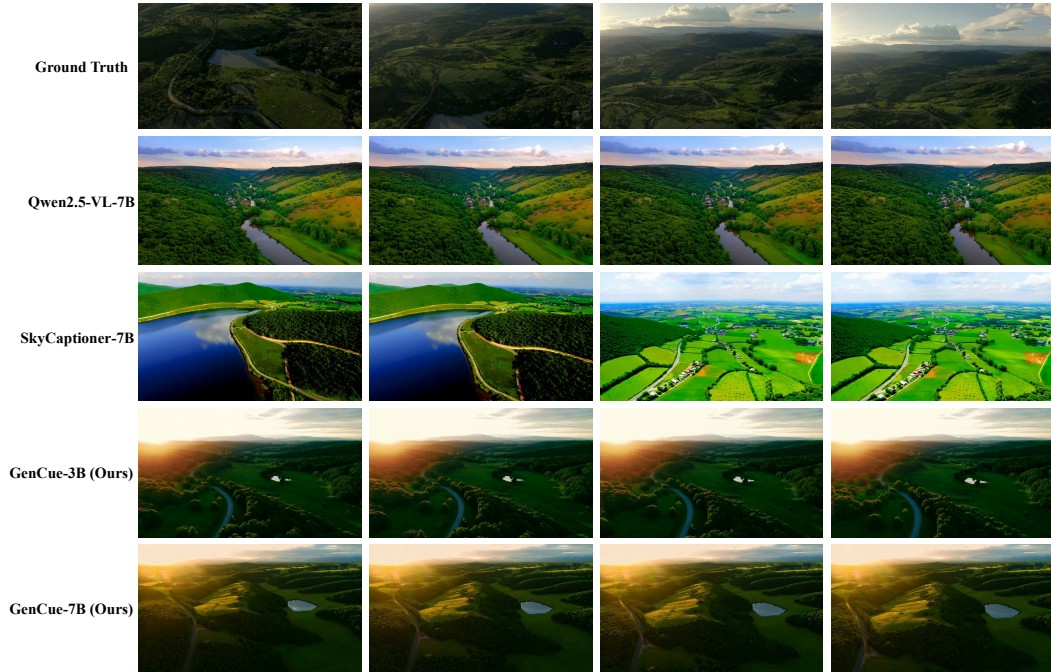

Figure 8: Comparison of reconstructed videos guided through captions generated by different captioning models. Note that our GenCue-3B and GenCue-7B successfully describes the color tone and the sunlight.

## C.2 EFFICIENCY EXPERIMENTS ON MORE GROUP SIZE SETTINGS

We conducted additional experiments to evaluate the efficiency of PrefixGrouper across a wider range of group sizes, with the results presented in Figure 9. The findings show that PrefixGrouper consistently achieves stable reductions in computational load and memory consumption across various context lengths and group sizes. This further demonstrates the effectiveness and generalizability of our proposed method.

## C.3 REWARD CURVE DURING TRAINING

Figure 10 presents the average reward curves over training steps for different reinforcement learning (RL) configurations.

## D DATA STATISTICS

We conducted a statistical analysis on the word count of our caption data and the number of verification points within our reinforcement learning (RL) data, with the results presented in Figure 11. The findings indicate that our captions and verification points contain highly granular information, demonstrating their ability to provide high-quality data. Furthermore, the video duration distribution can be found in Figure 12.

## E PROMPTS USED IN DATA GENERATION AND SCORE CALCULATION

The prompts used during the data construction process are shown in Figure 13 and Figure 14.

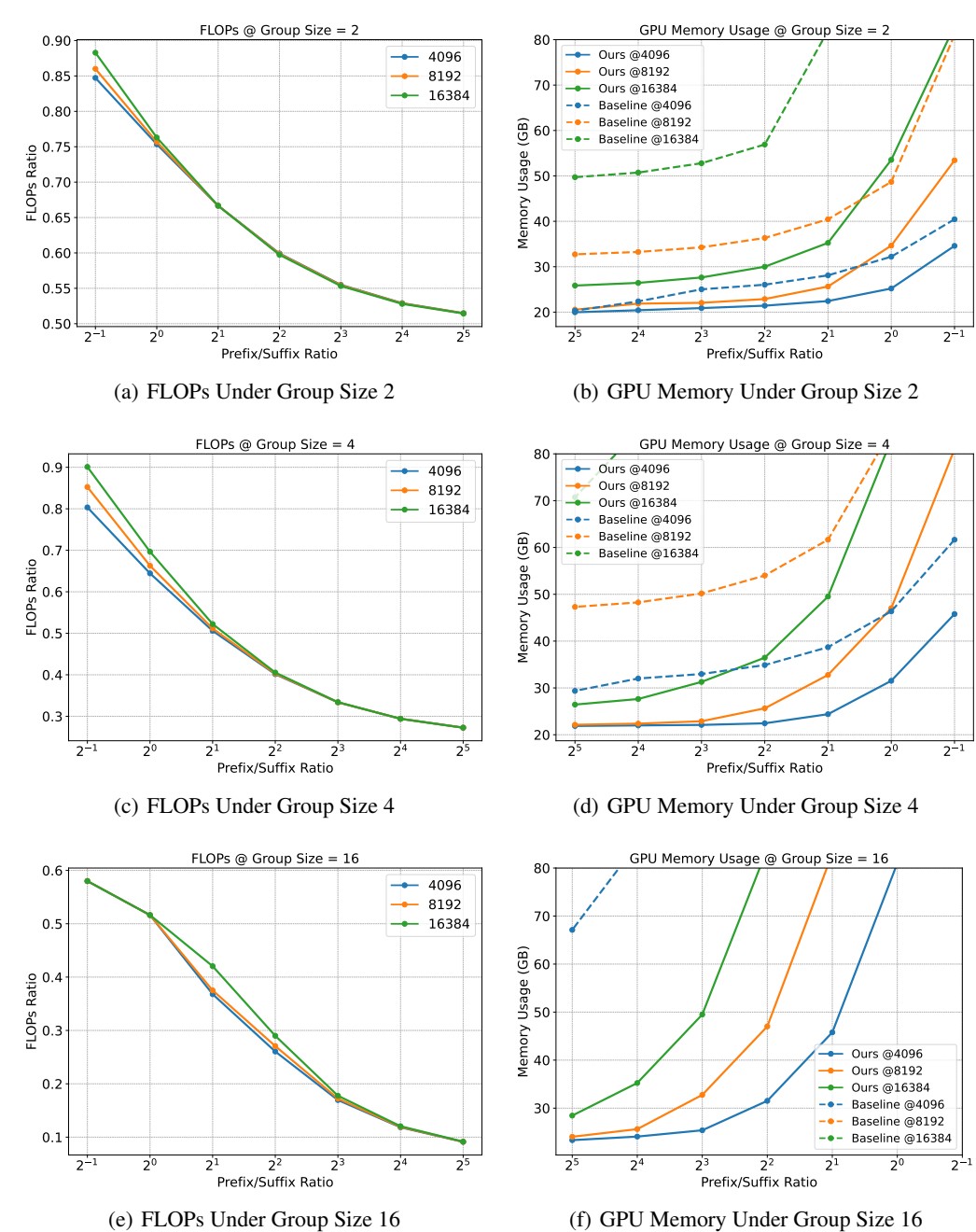

Figure 9: FLOPs and GPU memory usage comparison under different group sizes.

## F    REPRODUCIBILITY STATEMENT

Implementation details, evaluation protocols, and dataset descriptions are provided in the main text and appendix. Complete proofs are also included in the appendix. The full data, source code, and models will be released upon acceptance.

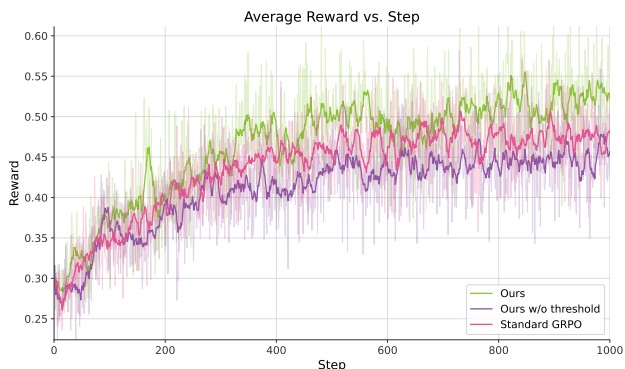

Figure 10: Average reward curves over training steps for different reinforcement learning (RL) configurations.

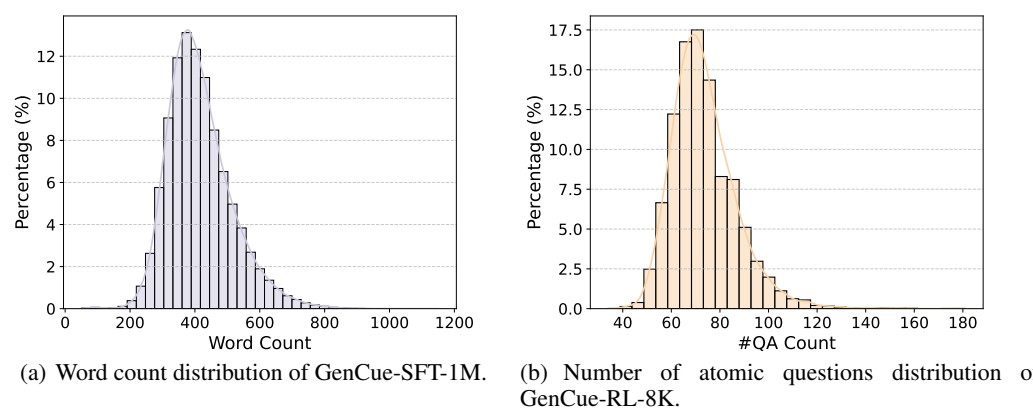

(a) Word count distribution of GenCue-SFT-1M.  (b) Number of atomic questions distribution of GenCue-RL-8K.

Figure 11: Word count distribution of GenCue-SFT-1M and number of questions distribution of GenCue-RL-8K.

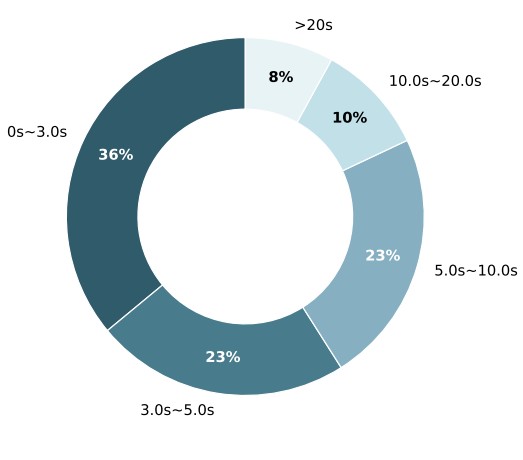

Figure 12: GenCue's video duration distribution.

**Generate an extremely detailed and granular description of the video. Consider the following aspects:**

**# Aspects**
**1. Composition: describe element arrangement (rule of thirds, symmetry, leading lines, etc.). Output plain text.**
**2. Color Scheme: specify dominant/palette colors, tone, etc. Output plain text.**
**3. Lighting: type, direction, color temperature & intensity, shadow characteristics, etc. Output plain text.**
**4. Cinematography: shot type, camera movement, shot angle, etc. Output plain text.**
**5. Visual Style: describe the style of the video. Output plain text.**
**6. Primary Subjects (only one or multiple): colors, shape & proportions, material/texture descriptors, pose/orientation, distinctive features, etc. Output a list where each item describes one subject in detail.**
**7. Background Elements: environmental context, etc. Output a list where each item describes one background element in detail.**
**8. Spatial Relationships: describe the relative positions of ALL subjects and background elements (covering ALL described items). Output a list where each item describes one relationship in detail.**
**9. Event Narration: describe the event in detail, including character/object interactions, collision/occlusion events, etc. Output a chronologically ordered list where each item describes a distinct event.**
**10. Motion Analysis: object motion, velocity description, etc. Output a chronologically ordered list where each item describes a distinct object motion.**
**11. Dynamic Changes: all the temporal changes of the above aspects. Output a chronologically ordered list where each item describes a distinct dynamic change.**
**12. Overall: Connect all the above content you have generated into a single paragraph. Output plain text.**

**# Expert Results**
**Here is information about the cinematography from an expert:**
**{expert_cinematography}**

**You should refer to the Expert Results in the response you generate.**

**# Instruction**
**1. Describe all observable details as granularly as possible, including but not limited to the listed aspects. Break down complex subjects and scenes into individual elements and describe them respectively.**
**2. Avoid using timestamps in your output.**
**3. Refer to the expert's results and your own judgement in the corresponding aspect. If any content is an empty string, supplement based on your own judgement.**
**4. When generating the final ``detailed`` field, you should be careful and MUST achieve 100% retention of the details of all previously generated content, which means that the content of ``detailed`` should be able to perfectly restore all the details of other aspects.**
**5. Use this JSON schema:**

**Caption = {{"composition": str, "color_scheme": str, "lighting": str, "cinematography": str, "style": str, "subjects": list[str], "background": list[str], "spatial_relationships": list[str], "event": list[str], "motion": list[str], "dynamic_changes": list[str], "overall": str}}**
**Return: Caption**

Figure 13: Prompts for structured caption generation.

**Here are detailed captions of a video:**

**```plain text**
**{detailed}**
**```**

**Here are 4 video caption aspects:**

**Aspects:**
**1. cinematography: shot type, camera movement, shot angle, shot position. Only focus on the above attributes of the lenses without mentioning the specific video content.**
**2. visual: Composition: element arrangement (rule of thirds, symmetry, leading lines, etc.); Color Scheme: dominant/palette colors, tone, etc.; Lighting: type, direction, color temperature & intensity, shadow characteristics, etc.; Visual Style: the style of the video.**
**3. frame: Subjects: colors, shape & proportions, material/texture descriptors, pose/orientation, distinctive features, etc.; Background Elements: environmental context, etc.; Spatial Relationships: the relative positions of ALL subjects and background elements (covering ALL described items).**
**4. temporal: Event Narration: the event in detail, including character/object interactions, collision/occlusion events, etc.; Motion Analysis: object motion, velocity description, etc.; Dynamic Changes: all the temporal changes.**

**Now you should first understand the video, and then extract questions according to the key points in the caption.**

**# Instruction**
**1. Answer: The questions should all be of Yes/No type, and the answers to them should always be "Yes", which means the key points in the questions are always based on the caption content.**
**2. Granularity: Be exhaustive. Break down the relevant information into the smallest possible details. Each question should inquire about a single, distinct atomic fact to avoid any redundant or overlapping questions.**
**3. Each question must be exclusively about ONLY ONE of the above 4 aspects.**

**Use this JSON schema:**

**Question = {{"question": str, "aspect": Literal["cinematography", "visual", "frame", "temporal"]}}**
**Return: list[Question]**

Figure 14: Prompts for atomic checklist generation.

## G    LLM Usage

Large Language Models (LLMs) were used to aid in the writing and polishing of the manuscript. Specifically, we used an LLM to assist in refining the language, improving readability, and ensuring clarity in various sections of the paper. The model helped with tasks such as sentence rephrasing, grammar checking, and enhancing the overall flow of the text. The LLM was also used as an experimental tool for data generation and for testing purposes. It is important to note that the LLM was not involved in the ideation, research methodology, or experimental design.

