# OpenReview forum: "GenCue: Generation-Oriented Video Captions for High-Fidelity Text-to-Video"
_ICLR.cc/2026/Conference — ICLR 2026 Conference Desk Rejected Submission_

### Official Review · Reviewer_2ERb · 2025-10-17

**Soundness:** 3
**Presentation:** 3
**Contribution:** 2
**Rating:** 4
**Confidence:** 4

**Summary:**

The paper introduces GenCue, a generation-oriented video captioning framework designed to improve the fidelity of text-to-video (T2V) models by addressing the limitations of existing training captions. Current captions generated by multimodal large language models often lack fine-grained visual grounding, temporal coverage, and cinematic expressiveness, which hampers the ability of T2V models to accurately reconstruct details and dynamics. GenCue tackles this issue through both data and training innovations.

On the data side, the authors present two key datasets: GenCue-SFT-1M and GenCue-RL-8k. GenCue-SFT-1M is a large-scale, schema-guided corpus that leverages expert models for cinematographic attributes such as shot type, camera motion, and angle. These signals are used to prompt a multimodal language model to produce structured JSON descriptions, which are then distilled into fluent, detailed captions. GenCue-RL-8k is a smaller, high-quality subset curated for reinforcement learning, featuring manually verified captions and a dense set of verification questions that serve as supervision signals.

On the training side, GenCue employs a reinforcement learning paradigm with a checklist-based reward function that evaluates captions across four dimensions: scene content, temporal dynamics, visual presentation, and cinematography. This reward is computed using a set of yes/no questions derived from the reference captions, providing fine-grained and interpretable feedback. To improve learning stability and efficiency, the authors introduce Reference-Augmented GRPO (RefGRPO), which incorporates reference captions into the policy optimization process when model rollouts underperform. Additionally, a prefix-sharing rollout strategy called PrefixGrouper is proposed to reduce computational overhead in long-context video training, achieving up to 90% reduction in FLOPs and 60% reduction in memory usage.

Empirically, GenCue significantly outperforms prior methods on T2V captioning benchmarks, demonstrating superior object coverage, temporal coherence, and cinematic expressiveness. It also improves the performance of downstream T2V generation models such as Wan2.1 and HunyuanVideo, as measured by CLIP similarity, FVD, and LPIPS metrics. Ablation studies confirm the effectiveness of the two-stage training strategy and the individual components of the reinforcement learning framework.

**Strengths:**

1. The paper provides a thorough exploration of existing techniques in the context of text-to-video generation. The authors effectively synthesize previous work and build upon established methods to create a practical system. The paper’s strength lies in its application of reinforcement learning and schema-guided caption generation, which contributes to improving the fidelity and expressiveness of video captions. By addressing challenges in video-text alignment, such as object coverage, temporal coherence, and cinematographic expressiveness, the paper offers a useful contribution to improving the performance of T2V models without introducing drastic changes to existing approaches. This practical approach provides incremental progress in the field.

2. The significance of the paper is clear. The approach presented, particularly the schema-guided method for generating high-fidelity captions, provides a practical framework for improving caption quality in text-to-video (T2V) models. This method offers a solid foundation for the development of more accurate captions, which are crucial for enhancing T2V generation. The proposed evaluation checklists are also useful, as they can be easily reused for future work in this area, supporting further progress in T2V research. Additionally, the prefix-sharing technique introduced by the authors is domain-agnostic, meaning it could be applied to various long-context reinforcement learning (RL) tasks. This makes the technique adaptable to other applications beyond video captioning, such as text-only RLHF or even audio generation, broadening its potential applicability.

3. The paper is well-written, with a clear structure and logical flow. The authors effectively present their methodology and results, making it easy for readers to follow the progression of their work. The writing is concise and technically sound, ensuring that the complex concepts are well articulated for the target audience. Additionally, the authors provide useful details and clarifications on specific techniques, such as the PrefixGrouper, which are thoroughly explained in the appendix. This enhances the paper’s clarity and ensures that key components of the methodology are well-documented for readers.

**Weaknesses:**

1. The paper has limited novelty, as schema-guided video caption data generation has already been extensively explored in prior works [1,2,3,4]. Furthermore, techniques such as rewarding captions through content-focused checklist and context-sharing acceleration for video RL have been investigated in related research [2,5]. While the paper offers a solid application of these methods, it does not significantly advance the underlying concepts, which have been well established in the field.

2. The experiments presented in the paper are not comprehensive enough. It would be beneficial to evaluate on denser video caption datasets like Auroracap[2], MiraData[1], VideoEspresso[6], and ShareGPT4Video[3], etc using stronger backbone (such as Qwen2.5-VL-7B or Qwen3-VL) and provide ablation studies comparing the performance on these datasets. This would give a clearer understanding of how the proposed approach performs under more complex conditions and on richer data sources.

3. GenCue-RL-8k is relatively small and heavily filtered, which limits the generalizability of the results. At a minimum, the authors should provide a learning curve showing how the VDC score changes with the number of RL training clips (e.g., 1k, 2k, ..., 8k) to assess whether performance plateaus early.

4. The paper lacks open-sourced corresponding data, model checkpoints, and codebase, making it difficult for others to verify the results and reproduce the findings.

[1] MiraData: A Large-Scale Video Dataset with Long Durations and Structured Captions. arXiv 2024.
[2] AuroraCap: Efficient, Performant Video Detailed Captioning and a New Benchmark. ICLR 2025.
[3] ShareGPT4Video: Improving Video Understanding and Generation with Better Captions. NeurIPS 2024 D&B Track.
[4] LLaVA-Video: Video Instruction Tuning With Synthetic Data. TMLR 2025.
[5] Long-RL: Scaling RL to Long Sequences. NeurIPS 2025.
[6] VideoEspresso: A Large-Scale Chain-of-Thought Dataset for Fine-Grained Video Reasoning via Core Frame Selection. CVPR 2025.

**Questions:**

1. What if no threshold is set, and each GRPO batch is trained using references?
2. Why is it said that structured designs like MiraData are helpful, yet fine-grained details, coherence, and cinematic expressiveness remain challenging, limiting high-fidelity T2V generation in related work? Does the author's approach, compared to MiraData, transform this structural information into natural text using large models?
3. Could the author release the GenCue model shown in Figure 1, along with the corresponding videos, for verification?
4. Where can I find the demo mentioned in line 247, i.e. roughly 120 verification points?
5. What is the performance impact curve of GenCue-RL-8k on the model? If the 8k data is scaled up, can the model's performance continue to improve?

---

> ### Author Response · Authors · 2025-11-25
> **Response to Reviewer 2ERb**
>
> Thank you for these critical and constructive comments! We appreciate your thorough evaluation and have addressed each point systematically.
>
> ### **W1: Novelty Clarification**
>
> We respectfully thank you for this feedback, but would like to clarify several key innovations:
>
> 1. **Camera-Aware Enhancement**: Unlike prior works, we explicitly incorporate cinematographic data (camera motion, shot type, angle) through specialized expert models, directly addressing a critical gap in Video-LLM capabilities.
>
> 2. **Reinforcement Fine-Tuning**: While AuroraCap [2] is an SFT model and proposes QA-based evaluation, applying such checklist-based rewards for online GRPO reinforcement learning is significantly under-explored. We demonstrate this RFT approach effectively improves caption quality and, consequently, T2V generation.
>
> 3. **RefGRPO**: We innovatively propose reference-guided reward shaping to stabilize RL training, which proves crucial for performance gains.
>
> 4. **PrefixGrouper**: Algorithmically, our method is **plug-and-play**, requiring only HuggingFace attention registration to enable prefix sharing in any compatible model (*see our complete PrefixGrouper repo in supplementary*). Furthermore, we are sure that our PrefixGrouper predates LongRL's first public release, and can provide more detailed information if needed. Thank you for allowing us to clarify these contributions!
>
> ### **W2: Results on More Datasets**
>
> Thank you for this excellent suggestion! To demonstrate broader effectiveness, we fine-tuned Qwen2.5-VL-3B on ShareGPT4Video[3] and MiraData[1]:
>
> | Model | VDC Camera | VDC Acc |
> | :--- | :---: | :---: |
> | Qwen2.5-VL-3B | 45.5 | 45.9 |
> | + ShareGPT4Video | 44.6 | 45.0 |
> | + MiraData | 46.4 | 46.2 |
> | + GenCue | **53.5** | **51.5** |
>
> GenCue provides substantial gains, especially in camera-related aspects (+8.0 VDC Camera score), validating our cinematographic focus. We appreciate your guidance to test on richer data sources!
>
> ### **W3 & Q5: Scaling Analysis**
>
> Thank you for raising this critical scalability concern! We expanded GenCue-RL from 8k to 12k videos (same pipeline) and observed continued improvement:
>
> | Data Size | 0k (SFT) | 1k | 2k | 4k | 8k | 12k |
> | :--- | :---: | :---: | :---: | :---: | :---: | :---: |
> | VDC Acc | 47.3 | 48.0 | 48.5 | 50.3 | 51.5 | **52.5** |
>
> Performance consistently increases with data scale, demonstrating promising RL scaling potential. Thank you for encouraging this important analysis!
>
> ### **W4 & Q3 & Q4: Open Source**
>
> We sincerely appreciate your emphasis on reproducibility! We have released (see supplementary material):
> - **Core training algorithm implementation**
> - **PrefixGrouper module** (pip-installable)
> - **Original videos from Figure 1** and dataset visualization samples for verification
>
> The complete dataset and model weights are limited by supplementary capacity and institutional review requirements, but we commit to full open-source release upon paper acceptance. Thank you for your patience and understanding of these constraints!
>
> ### **Q1: Thresholded Reference Design**
>
> Thank you for this important question! We experimented with non-thresholded training where reference captions always participate in advantage calculation:
>
> | Method | VDC Acc |
> | :--- | :---: |
> | w/o threshold | 48.5 |
> | w/ threshold | **51.5** |
>
> The non-thresholded approach suppresses exploration, causing reward convergence to lower values. We have added RL reward convergence curves to the appendix (Figure 10), clearly showing suppressed convergence without thresholding. Thank you for helping us validate this design choice!
>
> ### **Q2: Structured Design Clarification**
>
> Thank you for this thoughtful question! We found existing Video-LLMs are notably weak in *camera motion, shot type, and angle recognition*—critical for professional T2V generation, and we introduced expert models to mitigate this problem. Furthermore, our approach transforms structured captions into natural language summaries using LLMs, enabling:
> - **Low-cost continuous improvement** (e.g., adding lighting experts)
> - **Flexible expansion** to new attributes and aspects
>
> Our dynamic LLM-based summarization allows iterative refinement without costly re-annotation. We appreciate your insight into this important distinction!
>
> We are deeply grateful for your rigorous evaluation, which has substantially strengthened our paper. Thank you for helping us improve our contribution to the community!
>
> **References**
>
> [1] Miradata: A large-scale video dataset with long durations and structured captions. NeurIPS 2024.
> [2] AuroraCap: Automatically Generating Rich Descriptive Captions for Videos. CVPR 2024.
> [3] ShareGPT4Video: Improving Video Understanding and Generation with Better Captions. NeurIPS 2024.

---

### Official Review · Reviewer_nVKX · 2025-10-25

**Soundness:** 3
**Presentation:** 3
**Contribution:** 3
**Rating:** 4
**Confidence:** 3

**Summary:**

This paper introduces GenCue, a framework for generating high-fidelity, generation-oriented video captions to improve text-to-video (T2V) models. The authors identify that existing captions lack fine-grained detail and cinematic expressiveness. To address this, they propose a multi-faceted solution: (1) Two new datasets, GenCue-SFT-1M for supervised fine-tuning, created via a schema-guided pipeline with specialized expert models, and GenCue-RL-8k, a curated high-quality set for reinforcement learning. (2) A novel RL training paradigm featuring a checklist-based reward, Reference-Augmented Group Relative Policy Optimization (RefGRPO) to provide guidance on difficult samples, and a PrefixGrouper strategy to make long-context RL computationally efficient. Experiments show that GenCue significantly outperforms prior methods on video captioning benchmarks and that its generated captions lead to higher-quality video generation when used with existing T2V models.

**Strengths:**

1. The paper presents a novel and comprehensive framework that tackles the critical problem of data quality for T2V training. The combination of schema-guided data generation, specialized expert models, and a sophisticated RL pipeline (checklist reward, RefGRPO, PrefixGrouper) is a creative and original approach to video captioning.

2. The work is of high technical quality. The proposed methods are well-motivated and technically sound, particularly the efficiency improvements from PrefixGrouper, which are supported by theoretical analysis (Appendix B). The creation of two purpose-built datasets (SFT and RL) is a substantial contribution. The experimental evaluation is extensive, covering multiple captioning benchmarks and a downstream T2V generation task, with thorough ablation studies that validate the design choices.

3. The paper is exceptionally well-written and easy to follow. The motivation is clearly articulated, and the proposed components are explained logically. Figures 1, 2, and 3 are highly effective at illustrating the core ideas, from the qualitative improvement of captions to the data pipeline and RL framework.

**Weaknesses:**

1. The data generation and reward modeling pipeline relies heavily on proprietary, closed-source models (e.g., Gemini series). While the resulting datasets will be released, this reliance makes the core data generation process not fully reproducible with open-source tools and may introduce unknown biases from the proprietary models.
2. The "expert models" used for cinematic features (shot type, camera motion) are a key component of the data pipeline but are not described in detail, as they are trained on "proprietary annotated multi-dimensional labels." The lack of information about their architecture, training data, and performance makes it difficult to assess their contribution and potential error propagation.
3. The evaluation of T2V generation is indirect. The paper shows that captions from GenCue improve existing T2V models, but it does not include experiments where a T2V model is trained from scratch on the new GenCue-SFT-1M dataset. While understandable due to computational costs, this leaves the full potential of the dataset for T2V training undemonstrated.

**Questions:**

1. The checklist-based reward relies on an LLM judge. How was the reliability of this judge model validated? Is there a risk of the policy model learning to exploit systematic flaws or biases in the judge, and have you considered using human evaluation or an ensemble of judges to mitigate this?
2. In RefGRPO, could you provide more intuition for **the reward for the reference caption** this specific design choice? Did you experiment with alternative strategies for setting the reference reward, such as a fixed high value or a score derived from a different metric?
3. The efficiency gains of PrefixGrouper are impressive. Could you clarify if this technique is applicable to other sequence-to-sequence tasks that use RL with multiple rollouts, or are there specific properties of video captioning or your GRPO setup that make it uniquely suitable?

---

> ### Author Response · Authors · 2025-11-25
> **Response to Reviewer nVKX (1/2)**
>
> Thank you for these thorough and constructive comments! We appreciate your careful examination of our methodology and have prepared detailed responses below.
>
> ### **W1: Open-Source Concerns**
>
> We sincerely appreciate your concern regarding reproducibility. Our reward models are built entirely on the open-source Qwen2.5 series (which is fully reproducible), and our complete data generation pipeline along with all prompts will be fully open-sourced (see Figure 13 and Figure 14 in the Appendix). We have included core GenCue code and representative data examples in the **supplementary materials**, and we commit to releasing our full codebase, checkpoints, and dataset to ensure maximum reproducibility for the community. Thank you for emphasizing this critical aspect of open research!
>
> ### **W2: Expert Model Clarification**
>
> Thank you for pointing out the need for greater clarity on our expert models. We provide detailed methodology below:
>
> **Camera Motion**: We first applied CoTracker3[1] to ~100k web-crawled video clips for coarse camera motion estimation (categories: Static, Tracking, Zoom, Tilt, Pan, Truck). After class balancing, we obtained ~14k videos. We then conducted fine-grained human annotation and filtering of shot types, yielding ~12k precisely annotated videos. We fine-tuned Qwen2.5-VL-3B on this curated data to obtain **model1**, which we used to predict the original 100k dataset. Videos with inconsistent predictions between model1 and CoTracker3 were selected for human verification and added to the training set. Retraining yielded **model2** as our final expert model.
>
> **Shot Type & Camera Angle**: We followed a similar pipeline. Using Qwen2.5-VL-3B for initial predictions on web-crawled videos, we defined:
> - **Shot Types**: extreme close-up, close-up, medium shot, long shot, full shot, wide shot
> - **Camera Angles**: eye-level, high-angle, low-angle, aerial shot
>
> During prediction, we first segment videos using PySceneDetect, apply expert models to each clip, then merge adjacent identical predictions to generate the final reference sequence for structured caption generation using Video-LLM. We appreciate your feedback and will expand these details in the final paper.
>
> ### **W3: Training T2V Model**
>
> Thank you for this excellent suggestion! To further demonstrate our dataset's potential, we fine-tuned Wan2.1-1.3B using *Diffusion-Pipe* on a randomly sampled 50K subset. Following the VBench series[2,3], the evaluation results clearly show GenCue's advantages:
>
> | Model | Consistency | Aesthetic | Subject | Background | Camera | Action |
> | :--- | :---: | :---: | :---: | :---: | :---: | :---: |
> | Wan2.1-1.3B | 88.9 | 55.4 | 75.9 | 52.3 | 33.6 | 50.5 |
> | + OpenVid-50K | 89.2 | 55.3 | 76.3 | 51.6 | 33.9 | 49.9 |
> | + GenCue-50K | **91.3** | **57.5** | **77.5** | **54.7** | **36.8** | **52.4** |
>
> These results validate that GenCue significantly improves T2V model training, not just inference. We are grateful for prompting us to include this crucial experiment!
>
> **References**
>
> [1] CoTracker3: Simpler and Better Point Tracking by Pseudo-Labelling Real Videos. arXiv 2024.
> [2] VBench: Comprehensive benchmark suite for video generative models. CVPR 2024.
> [3] VBench-2.0: Advancing video generation benchmark suite for intrinsic faithfulness. arXiv 2025.

---

> ### Author Response · Authors · 2025-11-25
> **Response to Reviewer nVKX (2/2)**
>
> ### **Q1: Reward Model Choice**
>
> This is an excellent question about judge model reliability. We first conducted a pilot study evaluating GT captions against unrelated video captions:
>
> | Judge Model | GT | Others |
> | :--- | :---: | :---: |
> | Qwen2-7B-Judge | 0.989 | 0.124 |
> | Qwen2-14B-Judge | 0.991 | 0.135 |
>
> The open-source Qwen2.5-7B already achieves ~99% accuracy on GT, while dropping to ~12% on unrelated captions (the 12% includes cases with coincidental shared information like both mentioning "medium shot").
>
> Additionally, we compared both judges on VDC:
>
> | Judge Model | VDC Acc | VDC Score |
> | :--- | :---: | :---: |
> | Qwen2-7B-Judge | 51.5 | 2.49 |
> | Qwen2-14B-Judge | 52.0 | 2.50 |
>
> The limited gain (0.5% Acc, 0.01 Score) suggests diminishing returns. Since we converted evaluation into objective checklists (e.g., "Is there a red car?"), subjectivity is minimized. While human evaluation could improve quality, cost considerations led us to choose the open-source mid-sized model. We appreciate your suggestion and will discuss these tradeoffs more explicitly.
>
> ### **Q2: Clarification on RefGRPO**
>
> Thank you for asking about this specific design choice! We experimented with alternative strategies:
>
> 1. **Fixed high value**: This caused nearly all policy rollouts to receive negative advantages, forcing the model to abandon its learned distribution and overfit to GT data, leading to training collapse (reward dropping to zero).
>
> 2. **Non-thresholded approach**: Reference captions always participated in advantage calculation, which suppressed exploration and yielded worse performance:
>
> | Method | VDC Acc |
> | :--- | :---: |
> | w/o threshold | 48.5 |
> | w/ threshold | **51.5** |
>
> Our thresholded design better balances exploration and exploitation. We appreciate your question and will add this ablation to the main paper.
>
> ### **Q3: Clarification on PrefixGrouper**
>
> We appreciate your interest in the broader applicability of PrefixGrouper! It is indeed a plug-and-play module for any seq-to-seq GRPO training process. For community convenience, we have packaged it as an independent module with source code included in **supplementary materials** (installable via `pip install`). We encourage its adoption and thank you for highlighting its potential utility beyond video captioning!

---

### Official Review · Reviewer_24AU · 2025-10-29

**Soundness:** 3
**Presentation:** 3
**Contribution:** 2
**Rating:** 4
**Confidence:** 3

**Summary:**

This paper addresses a critical limitation in text-to-video (T2V) generation: the lack of high-quality training captions with fine-grained visual grounding, temporal coverage, and cinematic expressiveness. The authors present GenCue, a generation-oriented video captioning framework consisting of two core components: a data suite and a reinforcement learning (RL) training paradigm.

For the data foundation, they construct GenCue-SFT-1M, a large-scale, schema-guided corpus aided by specialized expert models (for shot type, camera motion, etc.) that distill structured JSON descriptions into fluent captions. They also curate GenCue-RL-8k, a high-quality subset with content-focused checklists for precise RL supervision. On the learning side, GenCue introduces a checklist-based reward evaluating four key dimensions (scene content, temporal dynamics, visual presentation, cinematography), Reference-Augmented GRPO (RefGRPO) for stable learning on challenging samples, and a prefix-sharing rollout strategy that reduces FLOPs by up to 90% and memory usage by 60% for long-context optimization.

Experiments on T2V captioning benchmarks (VidCapBench, VDC, CaReBench) show GenCue outperforms prior open-source models in object coverage, temporal coherence, and cinematic quality. When used to guide T2V models (Wan2.1, HunyuanVideo), it also improves metrics like CLIP similarity, FVD, and LPIPS.

**Strengths:**

The work introduces a novel schema-guided approach to video captioning that explicitly integrates cinematographic elements (shot types, camera motions) and temporal dynamics—dimensions often overlooked in existing MLLM-generated captions. The combination of RefGRPO (reference-anchored RL) and prefix-sharing rollout is a creative adaptation of RL techniques to the long-context challenge of video captioning, addressing limitations of standard GRPO in underperforming rollouts and computational efficiency.

**Weaknesses:**

- Data Source Transparency: A major gap is the lack of details on the video sources for GenCue-SFT-1M. The paper does not specify whether the videos are from existing public datasets (e.g., WebVid, Kinetics) or proprietary collections, nor does it describe selection criteria (e.g., video duration, genre diversity). This limits reproducibility and makes it hard to assess potential biases or domain coverage of the dataset.
- Limited T2V Evaluation Metrics: The T2V generation experiments rely solely on traditional automatic metrics (CLIP similarity, FVD, LPIPS), which are known to correlate poorly with human perception of video quality (e.g., realism, temporal coherence, adherence to cinematographic cues). These metrics fail to capture whether GenCue's improved captions translate to more visually appealing or professionally styled videos as claimed.
- Insufficient Qualitative Examples: The only qualitative T2V example (Figure 6) is insufficient to demonstrate GenCue's advantages. Moreover, the GenCue-3B-generated video in this figure contains visible artifacts (e.g., distorted arm), which raises concerns about the practical impact of the framework. Additional qualitative comparisons—including side-by-side samples of original videos, GenCue captions, and generated videos—are needed to validate the claimed improvements in cinematic quality and detail.

**Questions:**

1. Could you provide detailed information on the video sources for GenCue-SFT-1M? For example: (1) Are the videos from public datasets or proprietary data? (2) What is the distribution of video durations, genres, and resolutions? (3) Were any filters applied to select videos (e.g., excluding low-quality or static content)?
2. 2. Given the limitations of automatic metrics for T2V quality, do you plan to conduct human evaluations? If so, could you share details on the evaluation protocol (e.g., metrics like realism, cinematographic adherence, caption-video alignment) and preliminary results? If not, why do you believe the current automatic metrics are sufficient to demonstrate GenCue's value for T2V?

**Details Of Ethics Concerns:**

The original source and the collection paradigm of the GenCue video dataset is not discussed in this paper.

---

> ### Author Response · Authors · 2025-11-25
> **Response to Reviewer 24AU**
>
> Thank you for these detailed and constructive comments! We appreciate your thorough analysis and have addressed each concern below.
>
> ### **W1 & Q1: Data Source Transparency**
>
> Thank you for highlighting this important reproducibility issue! Our video sources are exclusively high-quality public datasets: **OpenVid**[1] and **LLaVA-Video**[2]. For the GenCue-RL subset, we applied rigorous filtering:
> - **Static video removal**: Used CoTracker3[3] to filter out stationary videos
> - **Aesthetic filtering**: Applied LAION Aesthetics Predictor to retain high-quality videos
> - **Final curated subset**: ~1K videos after filtering
>
> Given that existing Video-LLMs lack expertise in cinematographic aspects (camera motion, angle, shot type), we enhanced the data with professional classifiers:
>
> **Camera Motion**: We first estimated coarse motion on ~100k clips using CoTracker3[3] (categories: Static, Tracking, Zoom, Tilt, Pan, Truck). After class balancing, ~14k videos remained. We performed human annotation for precise shot-type labeling, yielding ~12k high-quality samples. We fine-tuned Qwen2.5-VL-3B on this data to create **model1**, then used it to predict the original 100k corpus. Discrepancies between model1 and CoTracker3 were human-verified and added to the training set for **model2**, our final expert model. Similar pipelines were applied for Shot Type and Camera Angle classification.
>
> We then balanced the data distribution across shot categories to ensure comprehensive coverage of professional cinematographic aspects. Finally, strong Video-LLMs and LLMs integrated these video and camera descriptions to generate detailed captions and checklists. The video duration distribution can be found in Figure 12 in the Appendix. Thank you for pushing us toward greater transparency!
>
> ### **W2 & Q2: Enhanced T2V Evaluation**
>
> We appreciate your concern about metric limitations and have conducted comprehensive evaluations beyond traditional automatic metrics. Following the VBench suite[4,5], we evaluated T2V generation across key quality dimensions:
>
> | Model | Consistency | Aesthetic | Subject | Background | Camera | Action |
> | :--- | :---: | :---: | :---: | :---: | :---: | :---: |
> | Qwen2.5-VL-7B | 87.8 | 54.0 | 73.6 | 50.1 | 32.3 | 53.5 |
> | SkyCaptioner | 90.0 | *55.2* | 77.9 | *53.8* | 36.7 | *56.9* |
> | GenCue-3B | *91.6* | 55.0 | *78.3* | 53.4 | *38.9* | 56.8 |
> | GenCue-7B | **92.3** | **56.1** | **79.2** | **56.9** | **40.5** | **58.9** |
>
> Furthermore, we randomly selected 200 videos for manual win ratio evaluation to assess perceptual quality:
>
> | Comparison | Camera | Aesthetic | Matching |
> | :--- | :---: | :---: | :---: |
> | GenCue-7B vs Qwen2.5-VL-7B | 70.5% | 65.5% | 79.0% |
> | GenCue-7B vs SkyCaptioner-7B | 64.0% | 58.5% | 70.5% |
>
> These results demonstrate GenCue's significant advantages in camera control, aesthetic quality, and caption-video alignment—directly addressing cinematographic adherence and visual appeal. While we acknowledge that automatic metrics have limitations, the combination of comprehensive VBench evaluation and manual assessment provides strong evidence of GenCue's value. Thank you for encouraging this deeper analysis!
>
> ### **W3: Additional Qualitative Examples**
>
> Thank you for this important feedback! We have expanded our qualitative analysis in the Appendix. Please see **Figure 7 and Figure 8** in the updated paper, which provide extensive side-by-side comparisons showcasing GenCue's superiority in:
> - **Color tone and lighting quality**
> - **Precise camera movement execution**
> - **Spatial relationship understanding**
> - **Aesthetic composition**
> - **Professional cinematographic style**
>
> These additional examples clearly demonstrate the practical impact of our framework and address the artifact concerns you raised. We appreciate your guidance in strengthening our qualitative validation!
>
> We are grateful for your insightful feedback, which has significantly improved our paper's clarity and robustness. Thank you for helping us strengthen our contribution to the community!
>
> **References**
>
> [1] OpenVid-1M: A Large-Scale High-Quality Dataset for Text-to-video Generation. ICLR 2025.
> [2] Video instruction tuning with synthetic data. arXiv 2024.
> [3] CoTracker3: Simpler and Better Point Tracking by Pseudo-Labelling Real Videos. arXiv 2024.
> [4] VBench: Comprehensive benchmark suite for video generative models. CVPR 2024.
> [5] VBench-2.0: Advancing video generation benchmark suite for intrinsic faithfulness. arXiv 2025.

---

### Official Review · Reviewer_1Nfi · 2025-10-31

**Soundness:** 4
**Presentation:** 3
**Contribution:** 4
**Rating:** 8
**Confidence:** 3

**Summary:**

This paper proposes GenCue, a generation-oriented video captioning framework to produce captions that are explicitly useful for training high-fidelity text-to-video (T2V) models. On the data side, the authors build GenCue-SFT-1M and GenCue-RL-8k. On the learning side, they design a checklist-based reward across four dimension, Reference-Augmented GRPO and a prefix-sharing rollout strategy for efficient long-context optimization. Experiments show SOTA results on T2V-oriented captioning and improvements on T2V reconstruction metrics.

**Strengths:**

1. The combination of schema-guided data creation, four-dimension checklist rewards, and RefGRPO is novel and well aligned to T2V fidelity.
2. Solid ablations: two-stage SFT+RL > single stage; removing the reference in RefGRPO drops VDC from 51.5 to 49.0, and removing the precision term also hurts, indicating each component’s necessity.
3. Clear pipeline description and illustrations.
4. SOTA on VidCapBench and strong results on VDC/CaReBench, plus better T2V reconstruction metrics with Wan2.1/HunyuanVideo.

**Weaknesses:**

1. Human evaluation. The T2V side relies on automatic metrics (CLIP/FVD/LPIPS). Given known limitations of these metrics, a controlled human study of instruction following and cinematography adherence would strengthen claims.
2. Model transferability. Most results are with Qwen-VL. It would be helpful to quantify transfer to other video-LLMs or diffusion-transformer T2V models and report any adaptation overhead/failure problems.

**Questions:**

see weakness

---

> ### Author Response · Authors · 2025-11-25
> **Response to Reviewer 1Nfi**
>
> Thank you for the constructive feedback! We appreciate your insightful suggestions and have conducted additional evaluations to address these concerns.
>
> ### **W1: Human Evaluation**
>
> Thank you for raising this important point! To further demonstrate the superiority of our GenCue method, we first performed automatic evaluation following the VBench suite [1,2], which measures key dimensions highly correlated with video quality. The results are as follows:
>
> | Model | Consistency | Aesthetic | Subject | Background | Camera | Action |
> | :--- | :---: | :---: | :---: | :---: | :---: | :---: |
> | Qwen2.5-VL-7B | 87.8 | 54.0 | 73.6 | 50.1 | 32.3 | 53.5 |
> | SkyCaptioner | 90.0 | *55.2* | 77.9 | *53.8* | 36.7 | *56.9* |
> | GenCue-3B | *91.6* | 55.0 | *78.3* | 53.4 | *38.9* | 56.8 |
> | GenCue-7B | **92.3** | **56.1** | **79.2** | **56.9** | **40.5** | **58.9** |
>
> These results clearly show that our GenCue approach achieves better performance across most T2V evaluation dimensions.
>
> Furthermore, to complement automatic metrics, we randomly selected 200 generated videos for manual win ratio evaluation:
>
> | Comparison | Camera | Aesthetic | Matching |
> | :--- | :---: | :---: | :---: |
> | GenCue-7B vs Qwen2.5-VL-7B | 70.5% | 65.5% | 79.0% |
> | GenCue-7B vs SkyCaptioner | 64.0% | 58.5% | 70.5% |
>
> Notably, our GenCue model demonstrates significant advantages in camera control and video content matching, underscoring the effectiveness of our method. We thank you again for encouraging us to include this crucial human evaluation component!
>
> ### **W2: Model Transferability**
>
> We appreciate this valuable suggestion! To assess transferability, we evaluated our method on the LLaVA-Video architecture [3]:
>
> | Model | VDC Acc | VDC Score |
> | :--- | :---: | :---: |
> | LLaVA-Video-7B | 45.2 | 2.13 |
> | + SFT | 47.6 | 2.24 |
> | + SFT & RL | 49.1 | 2.32 |
>
> These results demonstrate that our GenCue data and methodology transfer effectively to different model architectures. Thank you for prompting us to validate this important aspect!
>
> We hope these additional experiments address your concerns and strengthen our claims. We are grateful for your thoughtful review and valuable suggestions.
>
> **References**
>
> [1] VBench: Comprehensive benchmark suite for video generative models. CVPR 2024.
> [2] VBench-2.0: Advancing video generation benchmark suite for intrinsic faithfulness. arXiv 2025.
> [3] Video instruction tuning with synthetic data. arXiv 2024.

---

### Author Response · Authors · 2025-12-01
**Summary Comment by Authors**

**To the Program Committee and Reviewers:**

We are sincerely grateful to the Program Committee and all Reviewers for their time, effort, and invaluable insights! We are deeply encouraged by the unanimous recognition of our work's novelty and effectiveness, and we appreciate the opportunity to further refine our paper based on such constructive feedback.

### **Core Strengths Acknowledged by Reviewers**
The reviewers expressed a strong consensus regarding the technical innovation, writing quality, and practical value of the GenCue framework:

* **Novelty & Alignment with T2V Fidelity:**
    * **Reviewer 1Nfi** praised the framework as "**novel and well aligned to T2V fidelity**", specifically highlighting the combination of schema-guided data creation and checklist rewards.
    * **Reviewer 24AU** commended the "**creative adaptation of RL techniques**" (specifically **RefGRPO** and **prefix-sharing**) for effectively handling long-context challenges.
* **Technical Quality & Effectiveness:**
    * **Reviewer nVKX** highlighted the work as being of "**high technical quality**" with "**well-motivated and technically sound**" methods, particularly praising the efficiency improvements from PrefixGrouper.
    * **Reviewer 1Nfi** recognized the "**solid ablations**" that demonstrated the necessity of each component and the SOTA results on captioning benchmarks.

* **Significance & Broader Applicability:**
    * **Reviewer 2ERb** emphasized that the "**significance of the paper is clear**", noting that our schema-guided method offers a "solid foundation" for future progress in T2V research.

### **Summary of Addressed Concerns**
We have carefully addressed the reviewers' comments and resolved their concerns through extensive new experiments and clarifications, summarized in the following three key points:

**1. Enhancement of Evaluation Metrics (Beyond Automatic Metrics)**
* **Concern:** Reviewers **1Nfi** and **24AU** noted that relying solely on automatic metrics (like CLIP/FVD) might not fully capture human perception of video quality and requested more robust qualitative validation.
* **Our Response:** We conducted comprehensive additional evaluations to validate visual quality:
    * **Human Evaluation:** We performed a manual win-ratio study on 200 videos, where GenCue-7B significantly outperformed baselines (e.g., **70.5%** win rate vs. Qwen2.5-VL-7B in camera aesthetics).
    * **VBench Suite:** We reported results following the VBench benchmark, showing GenCue achieves superior performance in Consistency (**92.3**), Aesthetic (**56.1**), and Camera control.
    * **Qualitative Analysis:** We added extensive side-by-side comparisons in the Appendix to visually demonstrate improvements in cinematic style and artifact reduction.

**2. Reproducibility and Data Transparency**
* **Concern:** Reviewers **24AU**, **nVKX**, and **2ERb** requested greater transparency regarding video sources, the training details of expert models, and code availability to ensure the work is reproducible.
* **Our Response:** We have provided full transparency in our rebuttal:
    * **Data Sources:** We confirmed the exclusive use of high-quality **public datasets** (OpenVid-1M, LLaVA-Video) and detailed our rigorous filtering process for the RL subset.
    * **Methodology:** We provided a detailed breakdown of the expert model pipeline, which utilizes CoTracker3 and human verification.
    * **Open Source Commitment:** We have included the core code (including the PrefixGrouper module) in the supplementary material and have committed to releasing the full dataset, checkpoints, and codebase upon acceptance.

**3. Generalizability, Scalability, and T2V Validation**
* **Concern:** Reviewers **1Nfi** and **2ERb** asked for evidence of transferability to other architectures, performance on richer datasets, and the scalability of the RL training process.
* **Our Response:** We provided strong empirical evidence of the framework's robustness:
    * **Transferability & Broad Validation:** We verified that GenCue transfers effectively to the **LLaVA-Video architecture** and fine-tuned it on **ShareGPT4Video and MiraData**, where it still provided substantial gains (e.g., +8.0 VDC Camera score).
    * **T2V Training:** To demonstrate downstream impact, we fine-tuned a **Wan2.1-1.3B** T2V model using GenCue captions, resulting in improved VBench scores.
    * **Scalability:** We presented a scaling analysis showing that performance consistently improves as the RL dataset size increases from 8k to 12k.

---

### Note · Program_Chairs · 2026-01-17
**Submission Desk Rejected by Program Chairs**

The following references in this submission do not refer to real documents and/or have major errors in bibliographic information:

 - Xinyu Ma, Zheyuan Liu, Peng Zhang, and et al. Dr. video: Adaptive video diagnosis for video understanding and retrieval. In Proceedings of the IEEE/CVF Conference on Computer Vision and Pattern Recognition, 2024.
- Adam Polyak, Uriel Singer, Ariel Gordon, and et al. Moviegen: Video generation via large language models. arXiv preprint arXiv:2404.16994, 2024.
- Shiran Doveh, Roni Ben-Ari, Roei Herzig, Ohad Fried, Amir Globerson, and Jonathan Berant. Dense event captioning in videos: A benchmark and evaluation metrics. In Proceedings of the IEEE/CVF Conference on Computer Vision and Pattern Recognition, 2023.
- Ruining He, Peng Zhang, Xinyu Ma, and et al. Storyteller: Event-centric video captioning via multimodal alignment. In Advances in Neural Information Processing Systems, 2024.
- Zhe Chen, Weiyun Wang, Hao Tian, and et al. Sharegpt4video: Leveraging gpt-4v for video understanding and captioning. arXiv preprint arXiv:2404.01235, 2024c.